# Federated Behavioural Planes: Explaining the Evolution of Client Behaviour in Federated Learning

**Dario Fenoglio**
Università della Svizzera italiana
Lugano, Switzerland
`dario.fenoglio@usi.ch`

**Gabriele Dominici**
Università della Svizzera italiana
Lugano, Switzerland
`gabriele.dominici@usi.ch`

**Pietro Barbiero**
Università della Svizzera italiana
Lugano, Switzerland
`pietro.barbiero@usi.ch`

**Alberto Tonda**
INRAE
Paris, France
`alberto.tonda@inrae.fr`

**Martin Gjoreski**
Università della Svizzera italiana
Lugano, Switzerland
`martin.gjoreski@usi.ch`

**Marc Langheinrich**
Università della Svizzera italiana
Lugano, Switzerland
`marc.langheinrich@usi.ch`

## Abstract

Federated Learning (FL), a privacy-aware approach in distributed deep learning environments, enables many clients to collaboratively train a model without sharing sensitive data, thereby reducing privacy risks. However, enabling human trust and control over FL systems requires understanding the evolving behaviour of clients, whether beneficial or detrimental for the training, which still represents a key challenge in the current literature. To address this challenge, we introduce *Federated Behavioural Planes* (FBPs), a novel method to analyse, visualise, and explain the dynamics of FL systems, showing how clients behave under two different lenses: predictive performance (error behavioural space) and decision-making processes (counterfactual behavioural space). Our experiments demonstrate that FBPs provide informative trajectories describing the evolving states of clients and their contributions to the global model, thereby enabling the identification of clusters of clients with similar behaviours. Leveraging the patterns identified by FBPs, we propose a robust aggregation technique named *Federated Behavioural Shields* to detect malicious or noisy client models, thereby enhancing security and surpassing the efficacy of existing state-of-the-art FL defense mechanisms. Our code is publicly available on GitHub[1].

## 1 Introduction

Federated Learning (FL), a privacy-aware deep learning (DL) approach in distributed environments, is a dynamic system where many clients collaborate to train a model without sharing sensitive data, thus mitigating privacy risks [1, 2]. Analyzing the behaviour of FL systems is crucial to detect anomalies—such as distribution shifts [3, 4], biased data [5], or adversarial clients [6–9]—which may compromise the global model's predictive performance and introduce biases into its decision-

---

Author contributions are detailed in the Acknowledgements section.

[1] https://github.com/dariofenoglio98/CF_FL

38th Conference on Neural Information Processing Systems (NeurIPS 2024).

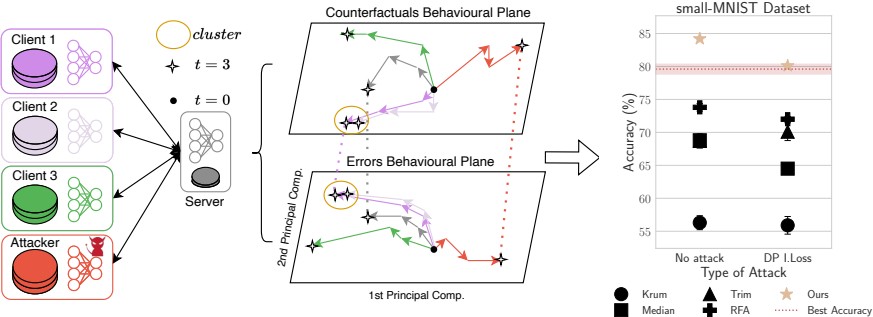

Figure 1: The Federated Behavioural Planes framework enables the visualization of client behaviour in FL from two perspectives: predictive performance (*Error Behavioural Plane*) and decision-making processes (*Counterfactuals Behavioural Plane*). It highlights client trajectories and similarities, offering insights into client interactions and supporting the introduction of a new and effective robust aggregation mechanism with performance that surpasses state-of-the-art baselines.

making process. Various strategies have been developed to detect FL anomalies [10–15]. However, existing techniques are not designed to track, visualise, and explain how client behaviours affect the performance of the global model, thus limiting human trust and control on FL dynamics.

Various studies have analysed models' behaviour in terms of predictive performance and decision-making processes independently. Predictive performance behaviour is primarily investigated in the context of non-linear optimisation using *behavioural spaces* [16–18]. This technique allows the visualisation of the predictive diversity of a set of regression models by considering the vector of errors that each model produces on a set of samples. The decision-making process is studied mainly in explainable AI (XAI) research using, for example, *counterfactual explanations* [19]. Counterfactuals can be used to identify relevant input features used by a model to make predictions, thus describing the position and orientation of the model's decision boundaries. However, these techniques have not yet been applied to FL systems, resulting in a lack of insight into how the behaviour of individual clients affects the overall model's accuracy and decision-making capabilities, leading to inefficiency in the training process.

To bridge this gap, we introduce *Federated Behavioural Planes* (FBPs), a method designed to visualise, explain, and give insights into the dynamics of FL systems. Our key innovation involves the creation of two behavioural planes for FL clients: one to highlight their predictive diversity and another to emphasise their decision-making process diversity via counterfactuals. Building on the client behaviour information provided by FBPs, to show their practical utility, we propose *Federated Behavioural Shields*, a robust aggregation mechanism that enhances security against malicious or noisy clients by accurately weighting the client models according to their constructive contributions during training. The results of our experiments demonstrate that: (i) counterfactual generators jointly trained with FL systems produce valid and client-specific counterfactual explanations which effectively describe clients' decision-making diversity; (ii) FBPs facilitate the identification of clusters of clients with similar behaviours (e.g., normal vs. outlier clients), allowing for tracking of their trajectories during the entire training; (iii) Federated Behavioural Shields surpasses existing state-of-the-art defense mechanisms, demonstrating that the information contained in FBPs provides valuable descriptors of client behaviour.

## 2 Background

**Federated learning.** FL systems [20, 3] involve a network of $K \in \mathcal{N}$ clients, coordinated by a central server, which collaboratively train a DL model. Each client $k$ possesses a local and private dataset characterised by a set of $z \in \mathcal{N}$ features $x^{(k)} \in X^{(k)} \subseteq \mathcal{R}^z$ and a set of $u \in \mathcal{N}$ class labels $y^{(k)} \in Y^{(k)} \subseteq \{0,1\}^u$. In each training round $t \in \mathcal{N}$, each client trains a model $f : X^{(k)} \rightarrow Y^{(k)}$ on local data to maximise the likelihood $\mathcal{L}(\theta^{(k)} \mid x^{(k)}, y^{(k)})$. Once trained, clients send their local model's parameters $\theta^{(k)}(t) \in \mathcal{R}^q, q \in \mathcal{N}$ to a central server which aggregates these parameters using a permutation-invariant aggregation $\oplus : \mathcal{R}^{q \times K} \rightarrow \mathcal{R}^q$ (such as the mean or median). The server then

sends the aggregated model parameters $\theta(t+1)$ back to the clients to start a new training round:

$$(\textit{local training}) \quad \theta^{(k)}(t) = \arg \max_{\theta^{(k)}(t)} \mathcal{L}(\theta^{(k)}(t) \mid x^{(k)}, y^{(k)}) \tag{1}$$

$$(\textit{aggregation}) \quad \theta(t+1) = \bigoplus_{k \in K} \theta^{(k)}(t) \tag{2}$$

While FL is an efficient process to safeguard privacy, the inherent lack of direct control over each individual client makes FL systems particularly vulnerable to various poisoning attacks [6–9, 21, 22, 14, 23–26]. These can be categorised into *model* and *data* poisoning attacks. Model poisoning involves altering gradients on compromised devices before transmission to the server [9, 14, 23, 24], while data poisoning indirectly manipulates gradients by tampering with training datasets on malicious devices [25, 27, 21].

**Counterfactual explanations.** Counterfactual explanations [19] describe a model's decision-making process by identifying minimal and plausible changes to an observed input's features that lead to a desired model prediction. In explainable AI, finding counterfactual explanations is framed as an optimisation problem where the objective is to identify, for each sample $x$, the nearest data point $x'$ such that the classifier $f(\theta, x')$ assigns a desired class label $y'$:

$$\arg \min_{x'} ||x - x'|| \quad \text{s.t.} \quad f(\theta, x') = y' \tag{3}$$

As a result, the variations in the input's features between $x$ and $x'$ offer actionable insights into how the model's decisions can be altered, highlighting the most important features.

**Semantic and behavioural spaces.** Semantic spaces [16, 17] represent the semantics of a model by considering the error vector $e = [\epsilon(f(\theta, x_i), y_i)]_{i=1,\ldots,n}$ that a model produces on a set of $n \in \mathcal{N}$ samples, where $\epsilon$ could be the Mean Squared Error, for instance. Given a set of $K$ models, the semantic space contains $n$-dimensional data points $[e_1, \ldots, e_K] \in \mathcal{R}^{K \times n}$. Behavioural spaces [18] summarise semantic spaces into lower-dimensional spaces applying a transformation $\psi_{n \to m} : \mathcal{R}^n \to \mathcal{R}^m$ with $m \ll n$ (typically $m = 2$ for most applications) i.e., $\psi_{n \to m}([e_1, \ldots, e_K])$.

## 3 Federated Behavioural Planes

**Problem definition:** Given an FL system composed of a set of $K$ clients with local data $(x^{(k)}, y^{(k)})$ and models $f(\theta^{(k)})$, we aim to analyse the evolution of the system to understand how clients impact the global model's predictive performance and decision-making over time. In Section 3.1, we introduce what drives our method and formalise the problem. In Section 3.2, we introduce Federated Behavioural Planes (FBPs), describing its components in more detail: the Error Behavioural Plane (Section 3.3) and the Counterfactual Behavioural Plane (Section 3.4). Finally, in Section 3.5, we introduce Federated Behavioural Shields, a new robust aggregation mechanism to enhance security in FL systems, showing a practical application of FBPs.

### 3.1 Dynamic behaviour of federated learning

FL is a dynamic process transitioning from a state where all participating clients behave randomly to a state where the behaviour of participating entities becomes coherent (the parameters of client models tend to converge to values which lead to high model accuracy). Thus, FL systems can be effectively analysed through the lens of dynamical systems using tools traditionally employed for such studies, such as differential equations. Defining $b(\theta^{(k)})$ as the evolving state of the behaviour of the client $k$ at time $t$, we introduce two primary forces that influence $b(\theta^{(k)})$: $g(\theta^{(k)}, x^{(k)}, y^{(k)})$, the local training dynamics, which drives the client towards its local optimum by leveraging information from the local dataset $(x^{(k)}, y^{(k)})$; and $b\left(\bigoplus_{k \in K} \theta^{(k)}\right) - b(\theta^{(k)})$, a correction term that periodically aligns $b(\theta^{(k)})$ with the aggregated state of all clients within the federated system. These dynamics can be encapsulated in the following differential equation[2], which describes how client behaviours evolve during training and are influenced by internal forces within the FL system:

---

[2]For simplicity, we omit the dependency on the time variable $t$ in our notation for variables $b$ and $\theta$.

$$\frac{\mathrm{d}b\left(\theta^{(k)}\right)}{\mathrm{d}t} = g\left(\theta^{(k)}, x^{(k)}, y^{(k)}\right)(1 - \delta_T) + \left[b\left(\bigoplus_{k \in K}\theta^{(k)}\right) - b\left(\theta^{(k)}\right)\right] \cdot \delta_T \qquad (4)$$

Here, $\delta_T$ is characterised by a periodic Dirac delta function, defined as $\delta_T = \sum_{r=0}^{\infty}\delta(t - r \cdot T)$, which triggers instantaneous adjustments at intervals determined by period $T$.

## 3.2 Federated Behavioural Planes (FBPs)

Instead of finding a general analytical solution for Equation 4 (which is not trivial and requires strong assumptions on its components), we aim to empirically analyse the phase space of a FL system by considering different descriptors of client behaviours. More specifically, we focus on investigating (i) *predictive performance*, evaluating how well the model is solving the task, and (ii) the *decision-making process*, as it contains information on how the model is solving the task. During each round, client behaviours are assessed through their respective models on the server, utilising a server-owned dataset reserved for this evaluation phase. This methodology aligns with existing protocols [9, 14, 28–32]. Each client behaviour is visualised through a two-dimensional plane: Error Behavioural Plane representing predictive performance and the Counterfactual Behavioural Plane to illustrate decision-making processes, collectively referred to as *Federated Behavioural Planes*. However, this framework offers a general approach to visualise and monitor different descriptors of the client behaviours simultaneously, which can be customised through specific functions, enabling the creation of additional planes.

## 3.3 Error Behavioural Plane (EBP)

To comprehensively evaluate each model from the predictive performance point of view, we analyse the errors made by the model on all samples, rather than relying solely on a simpler aggregate metric, such as loss or error [14]. This approach enables a more detailed examination of the differences in the model's performance as observed by Mouret and Clune [18]. Following the methodology proposed by Zhang et al. [33], we first construct a semantic error space for each model and then map it to a reduced space, called Error Behavioural Plane (EBP).

**Definition 3.1** (Error Behavioural Plane). Given a model $f$, parametrised with a set of weights $\theta^{(k)}(t)$, related to the client $k$ at round $t$, a dataset $(x^{(\text{server})}, y^{(\text{server})})$, owned by the server, and a dimensionality reduction technique $\psi_{n \to 2}$, the representation $e^{(k)}(t) \in \mathcal{R}^2$ in the EBP of the client $k$ is the following:

$$e^{(k)}(t) = \psi_{n \to 2}\left(\left[f(\theta^{(k)}(t), x_i^{(\text{server})}) - y_i^{(\text{server})}\right]_{i=1,\dots,n}\right) \qquad (5)$$

It is worth noting that two clients, despite having similar accuracy or loss, may receive significantly different representations in the EBP if they produce errors on distinct subsets of samples. In contrast, clients whose trajectories in the EBP converge over time form clusters representing clients whose predictive performance is similar on the same set of samples. However, clusters and trajectories in the EBP do not explain the decision-making process that leads to a prediction. This information could be used to further distinguish different types of clients and can be analysed using counterfactuals.

## 3.4 Counterfactual Behavioural Plane (CBP)

The analysis of a model's decision-making process is the main research objective of explainable AI. Counterfactual explanations represent one of the most effective techniques as they give insights concerning the position and orientation of decision boundaries. Similar counterfactual explanations indicate models having similar decision boundaries, i.e. models taking decisions using a similar decision-making process. To this end, most differentiable counterfactual generators are trained to model the training data distribution [34, 35], thus potentially providing insights on non identical distributions in clients' data. To produce these explanations, clients' predictive models should be concurrently trained with a counterfactual generator. Additional information on the optimisation objective are provided in Appendix A.2, A.3. To obtain the Counterfactual Behavioural Plane (CBP), we first compute the distances between the counterfactual distribution generated by the server using a

model of the client $k$ and the distribution of other clients. Then, we apply dimensionality reduction to obtain the CBP.

**Definition 3.2** (Counterfactual Behavioural Plane). Given a model $f$ with parameters $\theta^{(k)}(t)$, related to the client $k$ at round $t$, a set of samples $x^{(\text{server})} \in \mathcal{R}^{n \times z}$ owned by the server with $n$ samples and $z$ features, a distance function $d : \mathcal{R}^{n \times z} \to \mathcal{R}^+$, such as Wasserstein distance, and a dimensionality reduction technique $\psi_{K \to 2}$, the representation $c^{(k)}(t) \in \mathcal{R}^2$ in the Counterfactual Behavioural Plane of client $k$ in an FL settings is the following:

$$a^{(k)}(t) = \left[ \arg\min_{x_i'} ||x_i^{(\text{server})} - x_i'|| \text{ s.t. } f(\theta^{(k)}(t), x_i') \neq f(\theta^{(k)}(t), x_i^{(\text{server})}) \right]_{i=1,\ldots,n} \quad (6)$$

$$l^{(k)}(t) = \left[ d(a^{(k)}(t), a^{(i)}(t)) \right]_{i=1,\ldots,K}, \quad c^{(k)}(t) = \psi_{K \to 2}\left( l^{(k)}(t) \right) \quad (7)$$

CBP produces complementary information to EBP as clients which are similar in the CBP might be far away in the EBP (as discussed in Appendix B.4). Furthermore, as the purpose of CBP is to track clients' behaviour rather than explaining the model decision to a user, counterfactuals can be generated based on predictive models' embeddings, instead of input features, concealing sensitive information.

## 3.5 Federated Behavioural Shields – FBPs as a defence mechanism

FBPs provide descriptors of client behaviours during training, enabling various applications. Notably, trajectories in behavioural planes converging over time form clusters representing clients with similar predictive performance and decision-making process. This property can be used in practice to identify anomalies, such as malicious clients attempting to compromise FL training. In particular, leveraging this detailed information on client behaviours, we propose *Federated Behavioural Shields* (FBSs), a new class of robust aggregation strategies designed to enhance security in FL without requiring prior knowledge on the attack. This defensive mechanism generates a behavioural score in round $t$ for a client $k$, denoted as $s^{(k)}(t)$, which is formulated through the composition of multiple scores $s_j^{(k)}$ computed on $S$ behavioural spaces, to guide the aggregation process in creating the next round's global model $\theta(t+1)$, as outlined below:

$$s^{(k)}(t) = \frac{\prod_{j \in S} s_j^{(k)}(t)}{\sum_{i \in K} \prod_{j \in S} s_j^{(i)}(t)}, \quad \theta(t+1) = \bigoplus_{k \in K} s^{(k)}(t)\theta^{(k)}(t) \quad (8)$$

Specifically, based on the FBPs we previously defined, we can compute these scores as follows:

$$s^{(k)}(t) = \frac{s_{\text{error}}^{(k)}(t)s_{\text{cf}}^{(k)}(t)}{\sum_{i \in K} s_{\text{error}}^{(i)}(t)s_{\text{cf}}^{(i)}(t)} \quad s_{\text{error}}^{(k)}(t) = 1 - min(||e^{(k)}(t)||, 1) \quad s_{\text{cf}}^{(k)}(t) = \frac{1}{\frac{1}{K}\sum_{i \in K} l_i^{(k)}(t)}$$

The error score $s_{\text{error}}^{(k)}(t)$ measures the distance between the client $k$ and the optimal point at the center of the plane, while the counterfactual score $s_{\text{cf}}^{(k)}$ measure the average distance between the distribution of counterfactual generated by a client and all the other clients. In addition, considering that honest clients may occasionally deviate from the norm but generally contribute positively, we introduce a moving average mechanism to track client behaviours (see Appendix A.2 for details).

## 4 Experiments

The preliminary goal of our experiments is to assess whether counterfactual generators can provide insights in an FL context without compromising the performance of the predictor. We then visualise FBPs to verify that they can reveal information about the behaviour of various clients through the error and counterfatual behavioural planes (EBP and CBP). Lastly, we analyse the effectiveness of Federated Behavioural Shields as a robust aggregation mechanism, demonstrating the utility of the information provided by FBPs. Our experiments aim to answer the following questions:

- **Counterfactuals in FL:** Does the integration of counterfactual generators impact clients' predictive performance? Do counterfactuals have the same quality in FL compared to centralised scenarios? Could counterfactual generators be adapted for each client? Answering these questions is an essential preliminary step to check whether counterfactuals can be used to generate FBPs.
- **Explaining FL training:** Can trajectories in FBPs describe the evolving client behaviours during the training phase? Is it possible to visually identify clusters of clients using FBPs?
- **Leveraging FBPs information:** Do FBPs provide sufficient detail to Federated Behavioural Shields to enhance the security of the FL training process against security attacks?

This section describes essential information about the experiments. Further details on model configuration, training setup, and computational cost are presented in Appendices A.2, A.4, and A.6, respectively.

## 4.1 Data & task setup

In our experiments, we utilise four datasets: a *Synthetic* dataset (tabular) we designed to have full control on clients' data distributions, and thus test our assumptions; the *Breast Cancer Wisconsin* [36] (tabular); the *Diabetes Health Indicator* [37] (tabular); *small-MNIST* [38] (image); and *small-CIFAR-10* [39] (image), reducing its size by 76% to increase task difficulty and highlight client differences in performance. For all the experiments with the small-MNIST dataset, our approach involves generating counterfactuals at a non-interpretable internal representation level of the model instead of the input space. To reflect the most realistic cross-silo scenario [40], we distribute the training data among various clients such as data are not independent and identically distributed (IID) [41]. This represents the most challenging scenario to detect malicious clients due to the significant variations even among benign clients. Further details on the datasets and non-IID implementation are provided in Appendix A.1. Additional experiments analysing setup characteristics such as window length, local epochs, server validation set size, and differences between non-IID and IID scenarios can be found in Appendix B.

## 4.2 Evaluation

**Metrics.** To determine the efficacy of counterfactuals generated through end-to-end training in FL, we measure several key metrics: task accuracy ($\uparrow$ – higher is better); counterfactual validity ($\uparrow$) [19], which checks if the counterfactuals' labels align with user-provided labels; proximity ($\downarrow$) [35], assessing the realism of counterfactuals by their closeness to the training data (distance between the counterfactual and the closest data point in the training set with the same label); and sparsity ($\downarrow$) [19], which quantifies the changes made to the input to generate the counterfactuals (number of features changed between the initial sample and the counterfactual). The latter is quantified using Euclidean distance, as counting the number of changes provides less insight on the generated counterfactuals. To evaluate the effectiveness of client-specific adaptation, we analyse the relative change in client proximity between global and client-specific models, expressed as $(P_{global} - P_{local})/P_{global}$. Finally, we measure the task accuracy ($\uparrow$) of the FL system under different attacks and defenses. All metrics are reported as the mean and standard error across five experimental runs with distinct parameter initialization.

**Baselines.** In our experiments, we compare Federated Behavioural Shields with the following state-of-the-art robust aggregation methods: *Median* [11], *Trimmed-mean* [11], *Krum* [12], and *RFA* [42]. Further information is provided in the Appendix A.7.

**Federated Attacks.** We focus on attacks with realistic assumptions and where additional information outside the typical FL scenario are not available [10, 14, 23]. We test the following attacks: *Label-flipping (Data Poisoning)* [26, 43], changes each sample's label to $1 - y$ (performed in the context of binary classification); *Inverted-loss (Data Poisoning)* [25], creates an update that maximises the loss on the local dataset; *Crafted-noise (Model Poisoning)* [44], adds noise $\mathcal{N}(0, \beta \cdot \sigma(w^t))$ to the previous global model $w^t$, where $\sigma(w^t)$ is the standard deviation of $w^t$ and $\beta$ is a scale factor set to 1.2; and *Inverted-gradient (Model Poisoning)* [24, 45], inverts the gradient derived from the server's previous update, misaligning it with the true gradient. Further details are provided in Appendix A.8.

Table 1: Performance comparison of our model, which includes a Predictor and Counterfactual Generator (CF), across various settings: Local Centralised (Local CL), Centralised Learning (CL), Federated Learning (FL), and FL with only the Predictor in a non-IID setting.

| Metric | Dataset | Local CL Predictor + CF | CL Predictor + CF | FL Predictor + CF | FL Predictor |
|---|---|---|---|---|---|
| **Accuracy (↑)** | Diabetes | $55.9 \pm 0.5\%$ | $75.0 \pm 0.2\%$ | $74.7 \pm 0.1\%$ | $74.2 \pm 0.1\%$ |
| | Breast Cancer | $86.9 \pm 0.7\%$ | $97.7 \pm 0.0\%$ | $98.4 \pm 0.1\%$ | $97.7 \pm 0.4\%$ |
| | Synthetic | $75.0 \pm 2.0\%$ | $99.4 \pm 0.2\%$ | $99.8 \pm 0.1\%$ | $99.9 \pm 0.1\%$ |
| **Validity (↑)** | Diabetes | $87.6 \pm 2.6\%$ | $99.9 \pm 0.1\%$ | $99.9 \pm 0.0\%$ | *N/A* |
| | Breast Cancer | $100.0 \pm 0.1\%$ | $100.0 \pm 0.0\%$ | $100.0 \pm 0.0\%$ | *N/A* |
| | Synthetic | $97.1 \pm 1.9\%$ | $100.0 \pm 0.0\%$ | $100.0 \pm 0.0\%$ | *N/A* |
| **Sparsity (↓)** | Diabetes | $45.4 \pm 2.1$ | $34.5 \pm 1.7$ | $37.1 \pm 1.2$ | *N/A* |
| | Breast Cancer | $1459 \pm 25$ | $1325 \pm 20$ | $1448 \pm 43$ | *N/A* |
| | Synthetic | $8.63 \pm 0.15$ | $6.24 \pm 0.22$ | $6.14 \pm 0.07$ | *N/A* |
| **Proximity (↓)** | Diabetes | $8.91 \pm 0.61$ | $5.45 \pm 0.40$ | $6.23 \pm 0.44$ | *N/A* |
| | Breast Cancer | $61.2 \pm 2.1$ | $70.1 \pm 1.8$ | $72.1 \pm 5.5$ | *N/A* |
| | Synthetic | $0.142 \pm 0.026$ | $0.091 \pm 0.003$ | $0.089 \pm 0.002$ | *N/A* |

# 5 Key Findings & Results

## 5.1 Counterfactuals in FL

**Integrating counterfactual generators in FL optimisation does not compromise predictive performance (Table 1).** We compared model accuracy across four settings: two centralised learning (CL) and two FL, using three different datasets under non-IID conditions. For FL experiments, we used the traditional FedAvg approach with two variations: predictor-only and predictor with CF generator. The results indicate that our model, which involves the concurrent training of both the predictor and the counterfactual generator, achieves performance comparable to that of the predictor alone in FL. In the context of our model, both Local CL—where each client trains a model on its local data—and FL, implemented across all clients, comply with privacy standards [2]. For Local CL, we report the average accuracy of models trained independently by each client and evaluated on a common test set. However, only FL reaches the performance levels of the CL scenario, which assumes local access to all client data. This result indicates that incorporating counterfactuals during training does not compromise the predictor's performance, thus supporting their beneficial application without adverse effects on performance.

**Counterfactuals generated in FL have similar quality to those in CL (Table 1).** Unlike Local CL, FL leverages information from all clients, thereby producing counterfactuals that more accurately reflect non-IID clients' data. As shown in Table 1, FL achieves higher validity compared to the Local CL approach, indicating that the FL's counterfactuals better match ground-truth labels. Additionally, FL exhibits lower sparsity, which measures the number of modifications needed to achieve the counterfactuals. Table 1 shows these results using a counterfactual generator we adapted for this scenario (see Appendix A.3), however, similar conclusions are also obtained using out-of-the-box generators [34] (see Appendix B.2).

**Counterfactual generators can be adapted to specific clients (Figure 2).** In our study, we explored the impact of client adaptation on counterfactual generation within FL environments, as detailed in Section 4.2. Client-personalisation is key whenever we need to extract client-specific information. This adaptation can be achieved by training (or fine-tuning) a counterfactual generator on local client's data (which naturally happens at each round). The effectiveness of client-specific adaptation can be measured by the relative change in client proximity between global and client-specific models, as shown in Figure 2. The figure shows a marked reduction in relative proximity across the three datasets under non-IID conditions (up to 70% in the Synthetic dataset).

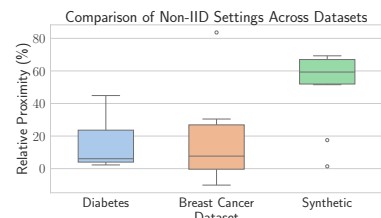

Figure 2: Relative variation of client-proximity across datasets.

This suggests that client-specific counterfactuals after adaptation are more representative of individual client datasets, providing unique descriptors of client-specific behaviours in the CBP.

## 5.2 Explaining FL training

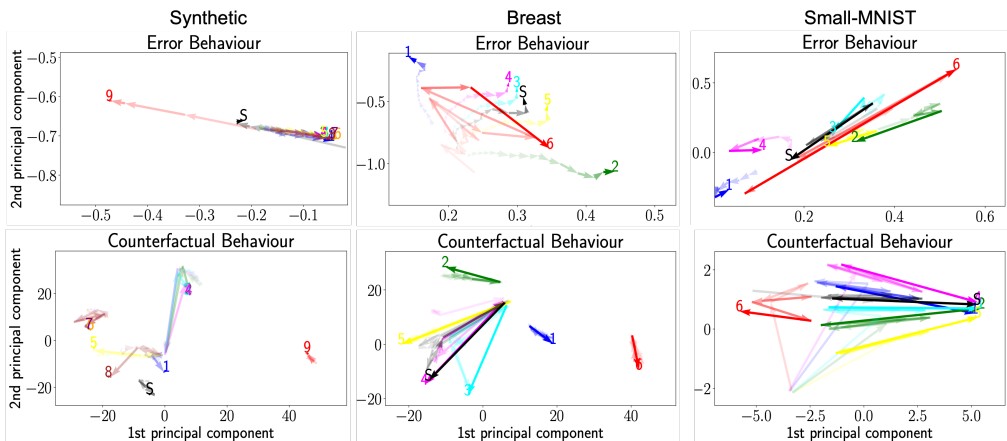

Figure 3: Client trajectories on Counterfactuals and Error Behavioural Planes for Synthetic, Breast Cancer, and small-MNIST datasets, corresponding to Inverted-loss, Crafted-noise, and Inverted-gradient attacks, respectively. The figure highlights the deviation of the malicious client (red) from honest clients, who tend to cluster together over time, along with the previous-round global model (S)

**The trajectories in FBPs enable the identification of different client behaviours during training (Figure 3).** Figure 3 illustrates FBPs' client trajectories over the last 15 rounds, where we introduced a different attack on each dataset (from left to right: Inverted-loss, Crafted-noise, and Inverted-gradient attacks). Each dataset was configured with five non-IID clients and one attacker (red). In the Synthetic dataset, to highlight the visualization of honest client clusters, the two largest clients were subdivided into three (Client 2,3,4) and two smaller clients (Client 6,7), respectively, forming distinct clusters. Specifically, on the EBP for the Synthetic dataset, the attacker (Client 9) significantly deviates from the trajectories of other clients, aiming to disrupt the server model (S). On the CBP, clusters with similar data distributions converge, indicating similarity in decision boundaries and client training data. In Breast Cancer and small-MNIST, FBPs are also able to give insights on the type of attack. On the Breast Cancer EBP, Client 6 exhibits a random trajectory around the server trajectory, which indicates a Crafted-noise attack. In small-MNIST, both the CBP and EBP reveal that at each round the trajectory of the attacker consistently moves in the opposite direction to that of the server, which indicates an Inverted-gradient attack. These insights might be useful for FL users as they explain the nature of clients' behaviours, allowing the identification of as various types of attacks. Additional visualizations are available in the Appendix B.3.

**FBPs allow the identification of clusters of clients (see Figure 3).** In the Synthetic dataset, CBPs's clusters reflect client-specific data distributions (Client 1,2-3-4,5,6-7,8 and Attacker 9). Clients sharing similar data distributions tend to cluster closely in the CBP, which in the Synthetic dataset is structured with adjacent slices in the feature space (for example, Client 1 is positioned between Client 8 and Client 2). Similarly, in Breast Cancer and small-MNIST, two primary clusters are identifiable in the CBP: honest clients and attackers. This clustering capability is crucial for users aiming to comprehend client characteristics without direct access to data or models' parameters, thereby informing strategic decisions in training a federated model. For instance, one might consider reducing the number of clients with identical data distributions to avoid redundancy and enhance training efficiency. Conversely, identifying distinct clusters in FBPs can be useful to maximise model performance (at the expense of the generalisation ability) by forming a Clustered FL system where independent federated models are trained using a subset of similar clients.

## 5.3 Leveraging FBPs information

**FBPs offer detailed insights into client behaviours, enabling the Federated Behavioural Shields to outperform existing state-of-the-art defense mechanisms (Figure 4).** Our comparative analysis demonstrates that Federated Behavioural Shields generally outperform traditional methods such as Krum, Median, and Trimmed-mean across various datasets including Breast, Diabetes, and small-

MNIST. The only exception was in the scenario of the Inverted gradient on the small-MNIST dataset, where Trimmed-mean performed better. Overall, our method enhanced the performance up to 10 percentage points (pp) over Median—the most robust aggregation baseline—when the FL system is under Label-flipping attack and up to 16 pp when the system is not under attack on small-MNIST. The proposed approach surpasses even a FedAvg aggregation in absence of attackers on the Breast Cancer and on the small-MNIST datasets, under both normal and crafted-noise conditions. These results suggest that an aggregation strategy based on predictive performance or decision-making similarities is superior to methods that solely consider sample count. Unlike other baselines, the independence of our method from prior knowledge about attackers establishes it as a robust aggregation tool in both adversarial and non-adversarial settings. Contrary to previous studies [14, 28], we find that predictive performance alone does not reliably identify malicious clients in non-IID settings, as honest clients may also show consistently low performance, leading to potential misclassification in the aggregation process. Occasionally, this reliance on predictive performance can reduce our method's accuracy compared to scenarios where only counterfactual information is used. For instance, under no-attack conditions, counterfactual information alone often offers more effective behavioural descriptors than combining it with predictive metrics. Further detailed analyses in Appendix B.12 indicate that using both descriptors generally yields better results than relying solely on counterfactual behaviour. Furthermore, visualizations in Appendix B.5 show how client scores consistently identify malicious clients during training, while Appendix B.11 demonstrates the robustness of our Federated Behavioural Shields against varying attack intensities.

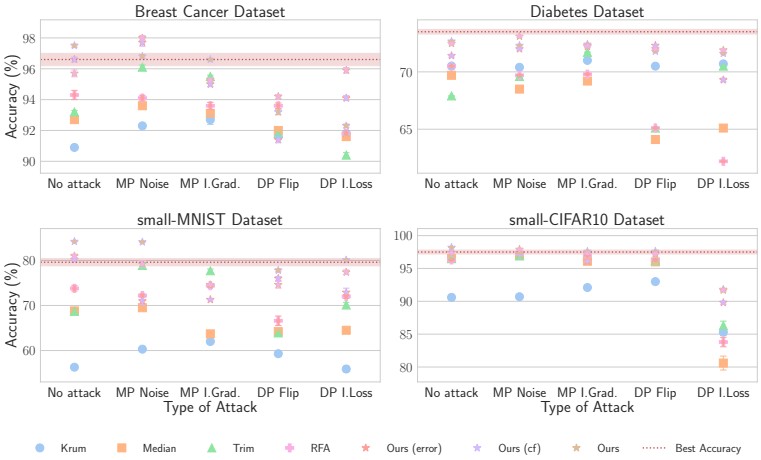

Figure 4: Comparative analysis of Federated Behavioural Shields and its simpler version with only counterfactuals (cf) or predictive-performance (error) versus Krum, Median, and Trimmed-mean defenses across five attack types—No attack, Crafted-noise, Inverted-gradient, Label-flipping, Inverted-loss—on three distinct datasets. Red dashed lines represent the accuracy achieved using FedAvg without attackers.

## 6 Discussion

### 6.1 Related works

Client behaviours in FL have been analysed using integrated visual tools [31, 46, 47] or indirectly through methods such as robust aggregation [10–14, 28, 30, 32, 40, 48–52] and clustering-based aggregation [53–59]. These studies typically focus on similarities in model or gradient parameters [10–13, 31, 40, 46–52, 55–59], under the premise that distinct client data distributions manifest as unique model parameters. Nevertheless, these methods might overlook valuable information about how client models behave with the data, particularly in terms of predictive performance. To mitigate this issue, previous studies adopted evaluation methods that utilise a clean validation set on the server to delineate clients' behaviour. Common descriptors used in such cases are straightforward metrics such as accuracy, error rates, and loss [14, 31, 46, 47, 53, 54]. Despite their utility, these metrics might not be able to reveal behavioural patterns, such as those involved in the decision-making processes of models, which may suggest subtle similarities or manipulations. Notably, Wang et al. [46] introduced

a post-hoc explainable approach, Grad-CAM [60], to explain client model behaviours during training. However, Grad-CAM is limited to CNN models and provides primarily qualitative visual insights, which cannot be easily automated and thus still require human intervention. Similarly, SentiNet [32] uses Grad-CAM to detect potentially malicious regions in input images, yet client behaviour assessments continue to rely primarily on evaluating predictive performance on manipulated and unmanipulated images. To the best of our knowledge, this work represents the first systematic attempt to formalise the evolving dynamics of clients in FL, showing how behavioural client trajectories affect the predictive performance and decision-making processes of the global model.

## 6.2 Limitations and future works

The primary constraint of our framework lies in its assumption that the server possesses a minimal validation set for querying client models. While this assumption is common across various methods, it can be mitigated, as demonstrated in Wang et al. [46], by generating synthetic data points. To this end, our approach might already integrate a potential mechanism as differentiable counterfactual methods can be used to generate synthetic data [61]. Furthermore, as indicated by the promising results in Appendix B.6, utilizing validation-independent descriptors, such as counterfactuals, renders our defense method robust against biased or unfair validation sets. Given the significance of ensuring fairness across clients, developing additional validation-independent descriptors represents a promising direction for future research. Another consideration is the computational overhead introduced by counterfactual generators, which, although minimal compared to other baselines (see Appendix A.6), is higher than that of traditional FedAvg. However, this overhead can be mitigated by using a smaller network for the counterfactual generator, thereby reducing the number of neurons (e.g., 1.8% of predictor parameters) without compromising accuracy.

Future work could leverage the extensive information provided by FBPs to explore additional strategies for optimizing the learning process, such as the development of Clustered FL among clusters of clients and the fine-grained categorization of attack types. Incorporating additional behavioural planes may also enhance the specificity of the FBPs explanations. Lastly, since our method allows for the integration of privacy-enhancing techniques such as Local Differential Privacy [62] and Homomorphic Encryption [63], future studies could analyze their impact on the overall performance and computational efficiency of our system.

## 6.3 Conclusions

In this work, we proposed Federated Behavioural Planes, a method to explain the dynamics of FL systems and client behaviours. This innovative method allows to visualise, track, and analyse client behaviours based on specific characteristics. Our focus was twofold: examining predictive performance by analysing prediction errors and investigating the decision-making process through counterfactual generation. The results of our experiments showed that Federated Behavioural Planes enable to track client behaviours over time, cluster similar clients, and identify clients' contributions to the global model with respect to a specific descriptor. Based on Federated Behavioural Planes' information, we introduced a novel robust aggregation mechanism that improves existing state-of-the-art methods by not requiring prior knowledge of the attacker. This work lays the foundation to explain the evolution of client behaviours, with the potential to enhance reliability and control over FL systems.

## Acknowledgments and Disclosure of Funding

This research was supported by the Swiss National Science Foundation, the European Union's Horizon Europe research program, and the Slovenian Research and Innovation Agency under SmartCHANGE (No. 101080965), TRUST-ME (No. 205121L_214991), and XAI-PAC (No. Z00P2_216405) projects.

**Author Contributions.** *Dario Fenoglio:* Conceptualization, Methodology, Implementation, Experiments, Writing – Original Draft, Writing and supplementary experiments – Rebuttal and Final Paper. *Gabriele Dominici:* Conceptualization, Methodology, Implementation of Counterfactual Generators, Writing – Original Draft. *Pietro Barbiero:* Conceptualization, Review, Supervision. *Alberto Tonda:* Review. *Martin Gjoreski:* Review, Supervision, Funding Acquisition. *Marc Langheinrich:* Review, Supervision, Funding Acquisition.

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

# A Experimental setup

## A.1 Datasets

In our experiments, we employ five distinct datasets:

- **Synthetic dataset** *(Tabular)*: It consists of two features randomly extracted from a range of -5 to +5. As shown in Figure 6.a, we partition this feature space into $K$ slices, where $K$ equals the number of clients. Each slice corresponds to a specific client. We assign a label of 1 to all data points where $x_1 > \alpha x_2$, effectively creating a linear decision boundary that varies with $\alpha$. For each client, we generally draw 1000 samples. This controlled setup allows us to precisely manipulate and visually understand the data distribution across different clients, as illustrated in Figure 6a.

- **Breast Cancer Wisconsin** *(Tabular)*: It contains data from 569 patients with 30 continuous variables derived from digitised images of a fine needle aspirate of a breast mass. The variables describe characteristics of the cell nuclei present in the image (e.g., radius, area, perimeter), aimed at predicting breast cancer [36].

- **Diabetes Health Indicator** *(Tabular)*: It comprises data from 70,692 patients, encompassing questionnaire-based variables (e.g., smoking, physical activity, fruit intake) and medical measurements (e.g., BMI, cholesterol levels), with a total of 21 features, to predict the presence of diabetes [37].

- **MNIST** *(Image)*: It is a comprehensive database of handwritten digits frequently used to benchmark image classification algorithms [38]. Each image is a 28x28 pixel grayscale representation of a digit, ranging from 0 to 9. To increase task difficulty and highlight client differences in performance, our experiments utilise only 10,000 of the available 70,000 images, referred to as small-MNIST. Additionally, we transform these images into color, randomly assigning colors with equal probabilities: red, green, or blue. To explore the decision-making process through counterfactuals, we employed a ResNet-18 architecture to extract 1,000 features from each image.

- **CIFAR-10** *(Image)*: The CIFAR-10 dataset comprises 60,000 color images categorised into 10 classes, widely used for evaluating object recognition algorithms [39]. Each image has a resolution of 32x32 pixels and is represented in RGB format. Similar to our small-MNIST setup, we reduce the sample size to 10,000 images. Features are extracted using a ResNet-18 architecture, resulting in 1,000-dimensional feature vectors for each image.

IID distributions across clients are achieved by randomly selecting samples from each dataset. In contrast, non-IID distributions are created using KMeans clustering [64] to form $K$ clusters of samples within each class—10 for the Synthetic dataset and 5 for others. We then assign the samples of the two nearest clusters from different classes to each client. To ensure each client's distribution is equally represented, we randomly extract 15% of the samples from each client and combine them into the test set to assess the performance of our experiments. We create a clean validation set on the server by extracting 89 samples for Breast Cancer and 250 samples for other datasets. Additionally, we partition each client's dataset locally, allocating 80% for training and 20% for validation.

## A.2 Model configuration

In our work, the model is composed of a predictor, any Deep Neural Network, which in our is a multi-layer perceptron that takes tabular data as input and makes a prediction. It is composed of five layers with hidden dimension equal to 512, 256, 256, 64, respectively. In addition, in our work, the model is also composed of a counterfactual generator, which can be implemented in different ways. The only requirement is that it can be trained concurrently with the main model, in an end-to-end fashion. To generalise our results, we tested the model with two different counterfactual generators. In Section 5, we used a counterfactual generator designed by us, adapting the intuition from Dominici et al. [65], that explicitly optimises the generation of counterfactual with respect to specific labels using two VAEs, named CFGen and described in Appendix A.3. On the other hand, in Appendix B.2 we also tested VCNet [34], which could generate counterfactual training a VAE, implementing it according to the original paper. In the context of creating Federated Behavioural Planes, PCA is employed as the dimensionality reduction technique $\psi_{n \to 2}$ within the EBP, centering the reduced

space around a zero vector, which signifies the absence of errors. For CBP, tSNE is utilised as the dimensionality reduction method $\psi_{K \to 2}$, ensuring the preservation of distances between clients and their respective clusters. Finally, in terms of the implementation of FBSs, at each round $t$, the score $s^{(k)}(t)$ for each client $k$ is computed using a moving average with a window length of $L$, as follow:

$$s^{(k)}(t) = \frac{1}{L} \sum_{i=0}^{L-1} s^{(k)}(t-i) \tag{9}$$

Additionally, client exclusion ($s^{(k)}(t) = 0$) is automatically triggered if a client exhibits a score approaching zero (i.e., $< 10^{-7}$) within the specified window.

### A.3 CFGen architecture

CFGen is designed to adapt the approach proposed by Dominici et al. [65] to tabular data, eliminating the need for predefined concepts. It is a latent variable model that generates counterfactuals through variational inference. To this end, we have two random variables $z$ and $z'$. These variables represent latent factors of variation whose probability distributions are easier to model and sample compared to those for $x$ and $x'$. We also include dependencies from $y$ to the counterfactual latent distribution $z'$ in order to explicitly model the dependency of $z'$ on the class labels, resulting in the following overall probabilistic graphical model:

$$\tag{10}$$

This way, the generative distribution factorises as:

$$p(\hat{x}, y, z, x', y', z', x) = p(\hat{x}, y|z)p(x', y'|z')p(x|z)p(\mathbf{z}|\hat{x}, y) \tag{11}$$

$$p(\hat{x}, y|z) = p(y|\hat{x})p(\hat{x}|z), \quad p(x', y'|z') = p(y'|x')p(x'|z'), \quad p(\mathbf{z}|\hat{x}, y) = p(z)p(z'|z, \hat{x}, y) \tag{12}$$

In our approach, $p(y|\hat{x})$ and $p(y'|x')$ are the task predictor; $p(\hat{x}|z)$ and $p(x'|z')$ are the same decoder. In practice, we assume that the input $x$ is always observed at test time, making the term and $p(x|z)$ irrelevant. Finally, $p(z)$ is a standard normal prior distribution and $p(z'|z, \hat{x}, y)$ is a learnable normal prior whose mean and variance are parametrised by a pair of neural networks $\phi_{p\mu}$ and $\phi_{p\sigma}$.

**Amortised inference.** CFGen amortise inference needed for training by introducing two approximate Gaussian posteriors $q(z|\hat{x})$ and $q(z'|z, \hat{x}, y, y')$ whose mean and variance are parametrised by a pair of neural networks $(\phi_\mu, \phi_\sigma)$ $((\phi_{\mu'}, \phi_{\sigma'})$, respectively).

**Optimization problem.** In order to obtain these counterfactuals, it is important to optimise their generation during training. CFGen is trained to maximise the log-likelihood of tuples $(\hat{x}, y, y')$, while observing $x$. Following a variational inference approach, we optimise the evidence lower bound of the log-likelihood, which results in the following objective function to maximise:

$$\mathcal{L} = \overbrace{\mathbb{E}_{z \sim q(z|x)}[\log p(\hat{x}|z)] + \log p(y|\hat{x})}^{\text{reconstruction of } \hat{x} \text{ and } y} - \overbrace{D_{KL}[q(z|x)||p(z)]}^{\text{prior regularization on } z}$$

$$+ \overbrace{\mathbb{E}_{z,z',x' \sim p(x'|z')q(z'|\alpha)q(z|x)}[\log p(y'|x')]}^{\text{reconstruction of } y'} - \overbrace{D_{KL}[q(z'|\alpha)||p(z'|z, \hat{x}, y)]}^{\text{prior regularization on } z'} \tag{13}$$

where $D_{KL}$ is the Kullback–Leibler divergence and $\alpha = (z, \hat{x}, y, y')$. Moreover, in order to enforce the counterfactuals to be as close as possible to the initial input, we add an additional term to the objective:

$$\mathcal{L}_{dz} = \overbrace{-D_{KL}[q(z|x)||q(z'|\alpha)]}^{\text{posterior distance}} - \overbrace{D_{KL}[p(z)||p(z'|z, \hat{x}, y)]}^{\text{prior distance}} \tag{14}$$

### A.4 Training configuration

Gradient Descent was employed as the optimisation algorithm, with a batch size equivalent to the dimension of the training dataset. Both the momentum and learning rate were set at 0.9 and 0.01,

respectively. For centralised training scenarios, the model was trained over 1,000 epochs. The Flower library was utilised to implement FL [66]. In all federated experiments, except for those evaluating local epochs, we employed 2 local epochs. During the assessment of various defense mechanisms, the number of communication rounds was capped at 200 and the window length for the moving average of 30 rounds. For comparisons with centralised training, 1,000 rounds were used. For client personalization, the generator was trained across 25 local epochs. In each experiments, performance metrics were evaluated on the model that exhibited the lowest aggregated loss during training. The aggregated loss represents the weighted average of client losses, evaluated on each client's local validation set, proportional to the respective number of samples.

## A.5 Code, licenses and hardware

For our experiments, we implement all baselines and methods in Python 3.9 and relied upon open-source libraries such as PyTorch 2.2 [67] (BSD license), Sklearn 1.4 [68] (BSD license), Flower 1.6 [66] (Apache License). In addition, we used Matplotlib [69] 3.8.2 (BSD license) and Seaborn [70] 0.13 (BSD license) to produce the plots shown in this paper. Data processing is performed using Pandas [71] 2.2 (BSD license). The four datasets we used are freely available on the web with licenses: Breast Cancer Wisconsin (CC BY-NC-SA 4.0 license), Diabetes Health Indicators (CC0 license), MNIST (GNU license), and CIFAR-10. Our code, along with all the necessary details to reproduce the experiments, is publicly available on GitHub [3] under the MIT license. Additionally, we provide pseudo-code for both client-side (Algorithm 2) and server-side (Algorithm 1) implementations of our proposed approach, which includes creating behavioural planes on the server and applying our FBSs. All experiments were conducted on a workstation equipped with an NVIDIA RTX A6000 GPU, two AMD EPYC 7513 32-Core processors, and 512 GB of RAM.

---

**Algorithm 1** The Federated Behavioural Shields Algorithm

---

**Require:** Initial model $f(\theta(0))$, number of communication rounds $T$, number of selected clients per round $M$, clean validation set $(x^{(server)}, y^{(server)})$, set of plane functions $S$
1: **for** $t = 0, 1, \ldots, T-1$ **do**
2:      Sample $M$ out of $N$ clients
3:      Send $\theta(t)$ to the selected $M$ clients
4:      **wait**
5:      **for** each selected client $k$ in $M$ **in parallel do**
6:          $x'^{(k)}, y^{(k)} = f(\theta^{(k)}(t+1), x^{(server)})$
7:      **end for**
8:      **for** $plane\_fn$ in $S$ **do**                          ▷ Creation of planes
9:          $s_j^k(t) = plane\_fn(x'^{(k)}, y^{(k)}, x'^{(k)}, y^{(server)})$      ▷ Our case: Eq.5 and Eq.6-7
10:     **end for**
11:     Behavioural planes visualization (FBPs)
12:     $s^{(k)}(t)$ = client score in Eq. 8 (part 1)
13:     $f(\theta(t+1))$ = Model aggregation in Eq. 8 (part 2) with $s^{(k)}(t)$      ▷ Eq. 2 for FBPs
14: **end for**

---

**Algorithm 2** Local Training on Client $k$

---

**Require:** Initial model architecture $f$, number of local epochs $E$, local dataset $(x^{(k)}, y^{(k)})$
1: Receive current global model parameters $\theta(t)$ from the server
2: Initialise client model with global parameters: $f(\theta^{(k)}(t)) \leftarrow f(\theta(t))$
3: Update local model $\theta^{(k)}(t+1)$ by training for $E$ epochs      ▷ Both predictor and generator
4: Send updated model parameters $\theta^{(k)}(t+1)$ back to the server

---

## A.6 Computational cost

This section outlines the computational costs associated with our proposed method, focusing on three primary components: local computation, communication overhead, and server-side computation.

---

[3]https://github.com/dariofenoglio98/CF_FL

**Local computation.** Our methods (both FBPs and FBSs) integrate a counterfactual generator with the original predictor to analyse decision-making and provide insights into the client's data distribution. For small neural networks, local computation is minimal compared to other costs in the FL framework, such as communication latency and synchronization. In these cases, the counterfactual generator can produce counterfactuals for the model input without impacting training efficiency. As the predictor size increases, as shown with MNIST and CIFAR-10, we can efficiently generate counterfactuals for intermediate layers using a relatively small number of neurons. For instance, in our experiments with small-MNIST and small-CIFAR-10, we used a ResNet-18, which has 12.42M parameters and requires 71.1M GFLOPs for inference with an input RGB image of size 28x28. As shown in Table 2, the generator with embedding size of 128 contributes only 5.1% of the operations compared to the predictor alone. Furthermore, by reducing the embedding size to 32, we maintained performance while reducing GFLOPs to just 2.7% of the ResNet-18.

**Communication overhead.** Similar to the predictor, the counterfactual generator must be transmitted to the server for evaluation and aggregation, thereby increasing the number of model parameters sent. However, the additional communication overhead is marginal compared to the size of the predictor. As illustrated in Table 2, the counterfactual generator with an embedding size of 32, which maintains high task performance, consists of only 1.8% of the parameters of the ResNet-18. Consequently, with 32-bit precision, our counterfactual requires 0.92 megabytes (MB) out of approximately 49.68 MB for the predictor.

**Server-side computation.** On the server, our methods involve evaluating client models' performances on a clean validation set and calculating the pair-wise distances of the generated counterfactuals between clients. As shown in Figure 13, the validation set size can be relatively small, e.g., 250 samples, which can be processed with a single forward pass of the model. Consequently, the models' evaluation on the server, which can also be parallelised, is negligible compared to the computational load of calculating pair-wise distances between client counterfactuals.

The primary computational bottleneck on the server is the pair-wise Wasserstein distance between client counterfactuals. To address this, we use the sliced Wasserstein distance implementation, which has a computational complexity of $O(m \log m)$, where $m$ is the number of supports, given by $m = \text{n\_samples} \times$ reduction dimension ($\psi_{K \to 2}$). In our implementation, $m = 250 \times 2 = 500$, and this operation is repeated for each unique pair of clients (i.e., binomial coefficient $\binom{n}{2}$), leading to a computational complexity of $O(n^2 \cdot m \log m)$ where $n$ is the number of selected clients.

Compared to Krum, which has a complexity of $O(n^2 d)$ with $d$ being the number of model parameters, our pair-wise operation is more efficient for neural networks with more than approximately 4480 parameters, which is likely the case in most practical applications.

**Overall computational cost.** We evaluated the overall computational cost of our defense mechanism in comparison to both the Krum algorithm and the traditional FedAvg algorithm, considering different network sizes and varying numbers of clients. In all experiments, a validation set of 250 samples was used. For tests involving an increasing number of clients, our method incorporated a counterfactual generator utilizing 7.6% of the predictor's parameters (in the worst-case scenario), as described in the original paper. Computational time was measured over 10 training rounds across 3 folds, and we reported the mean and standard error of the time per round. As shown in Figure 5, our method's computational cost scales more efficiently with both the number of model parameters and the number of clients when compared to Krum. Specifically, our method adds only an additional minute per round for 200 clients compared to FedAvg, while Krum introduces over 15 minutes per round. Importantly, our approach maintains the robustness and security benefits, demonstrating its efficiency and scalability in large-scale FL scenarios.

### A.7 Robust aggregation baselines

This section details the robust aggregation methods used as baselines in our experiments. These methods are designed to mitigate the effects of poisoning attacks by malicious clients and include:

- *Median*: This approach performs aggregation using the median of the client updates [11]. Compared to the mean, the median is less affected by outliers. Theoretical guarantees

Table 2: Comparison of embedding sizes of the counterfactual generator

| | Metrics | | Model Parameters | | | GFLOP | | |
|---|---|---|---|---|---|---|---|---|
| Embedding size | Accuracy | Validity | Pred.+CF | CF | Increase | Pred.+CF | CF | Increase |
| 128 (paper) | $86.0 \pm 0.3\%$ | $100.0 \pm 0.0\%$ | $13.36M$ | $0.94M$ | $7.6\%$ | $74.7M$ | $3.6M$ | $5.1\%$ |
| 64 | $85.6 \pm 0.4\%$ | $100.0 \pm 0.0\%$ | $12.88M$ | $0.47M$ | $3.8\%$ | $73.6M$ | $2.5M$ | $3.5\%$ |
| 32 | $87.6 \pm 0.5\%$ | $100.0 \pm 0.0\%$ | $12.65M$ | $0.23M$ | $1.8\%$ | $73.1M$ | $1.9M$ | $2.7\%$ |

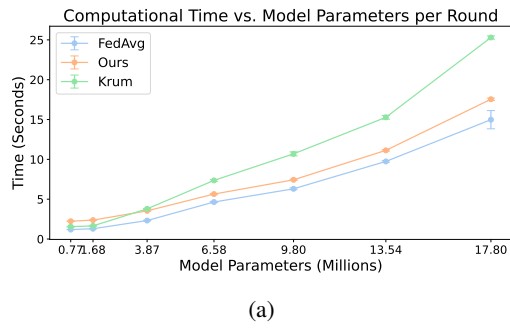

(a)

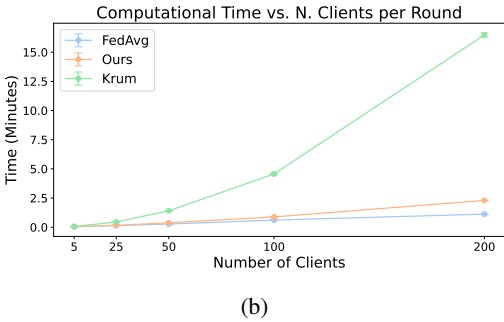

(b)

Figure 5: Comparison of the computational time of our proposed method against the traditional FedAvg, and the robust aggregation Krum per round of training, across (a) different model parameters and (b) different numbers of clients.

on the robustness of Median aggregation are provided in [11], while empirical evidence demonstrates that it exhibits better robustness than the more sophisticated Krum [14].

- *Trimmed-mean*: This approach [11] aggregates each dimension of input gradients separately. For each dimension $i$, it sorts the values of the $i^{th}$ dimension of all gradients, removes the $\beta_t$ largest and smallest values, and averages the remaining values to obtain the aggregate for dimension $i$. We use a default $\beta_t$ equal to 20% of the number of clients.

- *Krum*: Krum [12] is based on the intuition that malicious gradients must be far from benign gradients to poison the global model. Assuming knowledge of the upper bound on the number of malicious clients $m$, Krum selects the gradient from the set of $K$ input gradients that is closest to its $K - m - 2$ nearest neighbors in terms of the squared Euclidean norm.

- *Robust Federated Aggregation (RFA)*: RFA [42] replaces the standard arithmetic mean aggregation with the geometric median to ensure robustness against poisoned updates. It employs a Weiszfeld-type algorithm to compute the geometric median in a privacy-preserving and communication-efficient manner.

## A.8   Federated attacks

This section outlines and provides additional details about the federated attack scenarios implemented into our experiments, focusing on both model and data poisoning without prior knowledge of the server aggregation methods. We implemented these attacks assuming 20% of the clients are malicious:

- *Label-flipping (Data Poisoning)*: In a multiclass scenario, this attack changes all samples in the dataset with a source class $c_{src}$ into a target class $c_{target}$ [26, 43]. For binary classification, label flipping is implemented as $1 - c_{source}$, effectively inverting the class labels. For multiclass classification, we perform targeted label flipping by changing all instances of class 2 (Bird) to class 8 (Ship).

- *Inverted-loss (Data Poisoning)*: This attack aims to create an update that maximises the loss and, consequently, causes a significant drop in accuracy [25]. The implemented loss function follows the algorithm outlined in [25].

- *Crafted-noise (Model Poisoning)*: Inspired by the Free-riding attack described in [44], this attack aims for stealthiness by adding noise to the previous model received from the server,

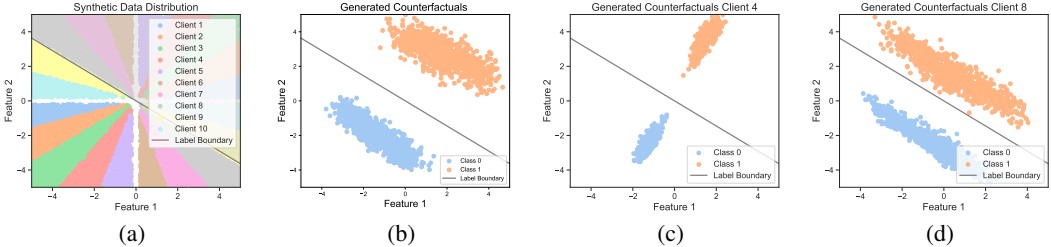

| (a) | (b) | (c) | (d) |

Figure 6: (a) Synthetic dataset. (b) Counterfactuals generated by the server on the synthetic test set. (c) Counterfactuals for Client 4 after adaptation. (d) Counterfactuals for Client 8, similar to (c). Notably, Client 4, with a data distribution perpendicular to the decision boundary, achieves effective adaptation, whereas Client 8 encounters more challenges.

$w^t$. The noise is added as follows: $w^* = w^t + \mathcal{N}(0, \beta \cdot \sigma(w^t))$, where $\sigma(w^t)$ represents the standard deviation of $w^t$ and $\beta$ is a scale factor. We adopted $\beta$ equal to 1.2, except when we tested the sensitivity values: 0.3, 0.5, 0.8, 1.2, and 1.6.

- *Inverted-gradient (Model Poisoning)*: This attack modifies the inner product between the true gradient and the malicious gradient updates sent by the attacker, affecting the alignment with the true gradient [24, 45]. In our implementation, the malicious gradient is inverted: $-\nabla w$. Given the lack of information about honest clients, the true gradient is taken from the previous update sent by the server, representing the previous improvement of the global model.

For both data poisoning attacks, the attacker uses a local dataset drawn from the same distribution as other clients and with an average number of samples.

## B  Additional experiments and analysis

### B.1  Client-specific adaptation in IID and non-IID scenario

As depicted in Figure 2, there is a significant reduction in the relative proximity measure between global and client-specific models across all three datasets on average, indicating a high degree of customization in the client models. Nonetheless, a few clients are still unable to tailor the model effectively to their specific distribution. To explore this phenomenon, we visually represent the generated counterfactuals pre-adaptation in Figure 6.a, showcasing models with the highest and lowest degrees of client-specific adaptation in Figures 6.c and 6.d, respectively. Notably, the most adapted model corresponds to client 4, whose data is almost perpendicular to the class boundary. In contrast, client 8, whose data distribution lies close to the boundary, struggles to adapt, remaining akin to the pre-adaptation conditions. This could highlight an increased capacity in personalisation in clients whose data is perpendicular to the learnt decision boundary, also indicating that personalization varies among clients, thereby offering more insights on the behaviour of the clients. Furthermore, we analyse the effects of client-specific adaptation under both IID and non-IID conditions. As shown in Figure 7, in contrast to non-IID, under IID conditions where data distributions are uniform between clients, the enhancements were minimal, indicating an alignment between global and local optima. The lack of customization implies that all client models share a similar decision-making processes, correctly reflecting with their IID condition. However, this setting also facilitates the identification of outliers or anomalous clients, which exhibit a higher degree of customization.

### B.2  Architecture independence in counterfactual generation

To explore the independence of FBPs from the counterfactual generator, we evaluate our methodology with an alternative model named VCNet [34], maintaining identical experimental conditions. As illustrated in Table 3, VCNet achieves comparable performance in both Federated Learning (FL) and Centralised Learning (CL) configurations, where data from all clients is placed on a single machine. Similarly to our counterfactual generator, the results demonstrate that the privacy-preserving training

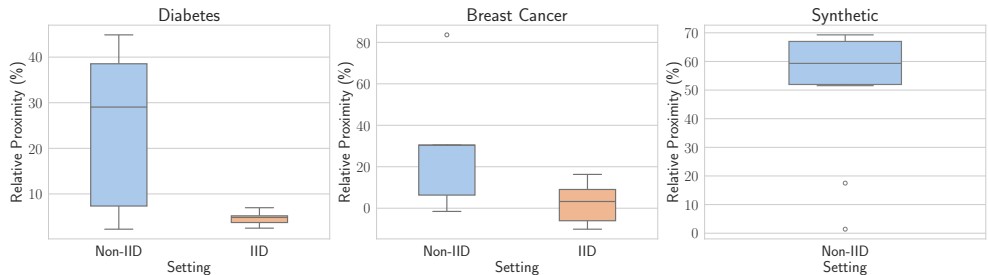

Figure 7: Relative variation of client-proximity across Diabetes, Breast Cancer, and Synthetic datasets in both IID and non-IID settings. Client-personalization is particularly effective in non-IID settings.

Table 3: Comparison of model performance in Local Centralised, Federated Learning, and Centralised Learning (i.e., privacy-intrusive) for non-IID setting using VCNet [34]

| Metric | Dataset | Local CL | CL | FL |
|---|---|---|---|---|
| **Accuracy (↑)** | Diabetes | 56.8±0.0% | 73.8±0.0% | 73.9±0.0% |
| | Breast Cancer | 84.0±0.2% | 97.0±0.3% | 97.5±0.6% |
| | Synthetic | 74.6±0.1% | 99.5±0.1% | 99.8±0.1% |
| **Validity (↑)** | Diabetes | 100±0% | 100±0% | 100±0% |
| | Breast Cancer | 100±0% | 100±0% | 100±0% |
| | Synthetic | 100±0% | 100±0% | 100±0% |
| **Sparsity (↓)** | Diabetes | 51.1±0.1 | 42.1±0.1 | 35.4±0.0 |
| | Breast Cancer | 2131± 10 | 1555±4 | 1560±19 |
| | Synthetic | 9.19±0.01 | 7.07±0.11 | 6.95±0.03 |
| **Proximity (↓)** | Diabetes | 11.58±0.28 | 9.23±0.24 | 8.17±0.56 |
| | Breast Cancer | 132.4±4.5 | 69.5±0.7 | 71.5±5.6 |
| | Synthetic | 0.080±0.002 | 0.096±0.006 | 0.090±0.002 |

strategy (i.e., FL) does not compromise the VCNet's effectiveness compared to the ideal CL condition. Operating under similar privacy constraints, FL enables VCNet to learn from the diverse distributions of client data, thereby surpassing the performance of the Local CL approach, where each client independently trains their own model.

To assess VCNet's capacity for adapting the counterfactual distribution to individual clients, we conduct client-specific adaptations starting from the global model achieved through FL. Figure 8 illustrates the relative variation in client-proximity across three datasets: Diabetes, Breast Cancer, and Synthetic. The observed significant reduction in proximity indicates that VCNet can be effectively personalised to each client's distribution, thereby confirming its utility in accurately reflecting client behaviours during the training process.

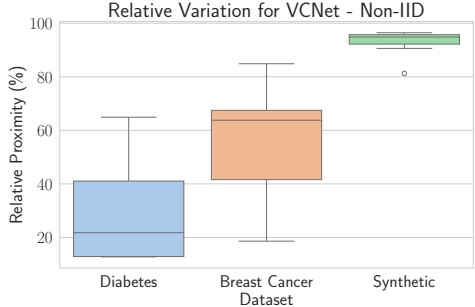

Figure 8: Relative variation of client-proximity across Diabetes, Breast Cancer, and Synthetic datasets for VCNET.

### B.3 Explaining FL training with Behavioural Planes

Figure 9 illustrates client trajectories within the FBPs for different scenarios from those presented in Section 5. Specifically, the left side of the figure displays the FBPs for the Synthetic dataset, now including an attacker employing an Inverted Loss attack, while the right side shows the FBPs for the Diabetes dataset under a Data Flip attack scenario. In both cases, the largest clients are subdivided into smaller clients (three and two), forming distinct clusters of clients (2,3,4 and 6,7 for Synthetic; 3,4,5 and 7,8 for Diabetes). These clusters are particularly evident on the CBP, and even within the Synthetic dataset's plane, one can observe clients with similar data distributions (e.g., Client 8's data distribution lies between that of Clients 7 and 1). This observation holds across all clients in the Synthetic dataset, underscoring the ability to highlight information about data distribution similarities among clients without direct access to their data. Moreover, Client 9 (the attacker) is noticeably diverging from the others across all four planes. It is crucial to note the distinct behavioural patterns of the attacker depending on the attack type. In the Inverted Loss scenario, Client 9 moves in the exact opposite direction to the others, converging at the same point in the EBP. Conversely, in the Diabetes dataset with a Data Flip attack, Client 9 simply diverges from the others, each moving towards different minima. This variation highlights the potential to identify the type of attack deployed by the malicious client based on these behavioural information.

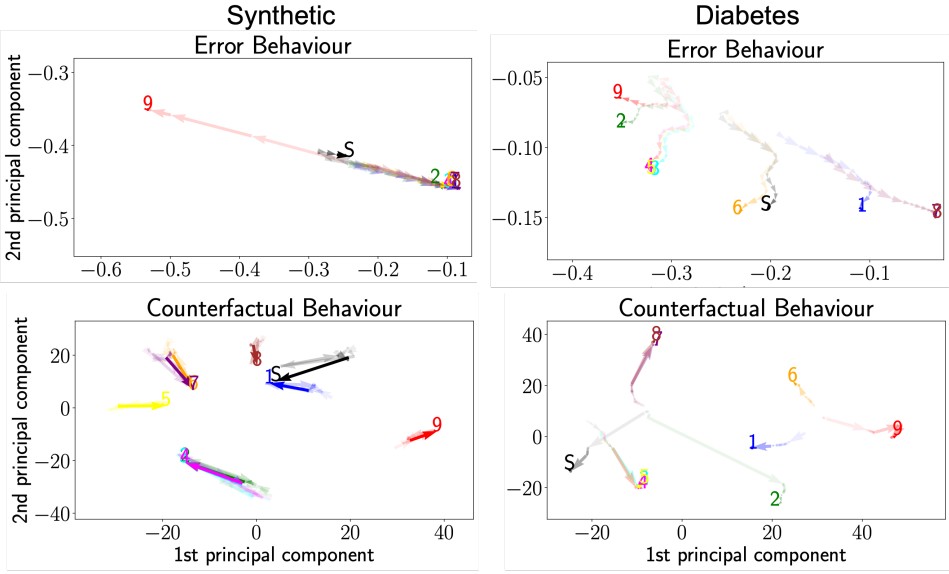

Figure 9: Client trajectories on Counterfactuals and Error Behavioural Planes for Synthetic, and Diabetes datasets, corresponding to Inverted-loss, and Label-Flipping attacks, respectively. The figure highlights the deviation of the malicious client (red) from honest clients, who tend to cluster together over time.

### B.4 The complementary roles of CBP and EBP

As previously observed in Figure 6, EBP and CBP provide orthogonal descriptions of client behaviours. In the Synthetic dataset, CBP effectively identifies clusters of clients with analogous data distributions. In contrast, EBP shows all honest clients moving in the same direction, thereby obscuring individual cluster distinctions but effectively contrasting the movements of honest clients with that of the attacker. Similarly, in the Breast Cancer dataset, the identification of an attack (i.e., Crafted-noise), is more discernible through EBP, which elucidates the direction of the malicious client relative to the server. These findings highlight the distinct yet complementary roles that EBP and CBP play in the analysis of client behaviour across different datasets.

To illustrate the complexity of model behaviour and the utility of EBP and CBP, consider the behaviour of three different models with respect to the same input changes. For instance, a data point with features [1, 2] is misclassified as class 1 by Model A and correctly classified as class 0 by Model B. However, both models would change their predictions to class 1 if the features were altered to [2, 3].

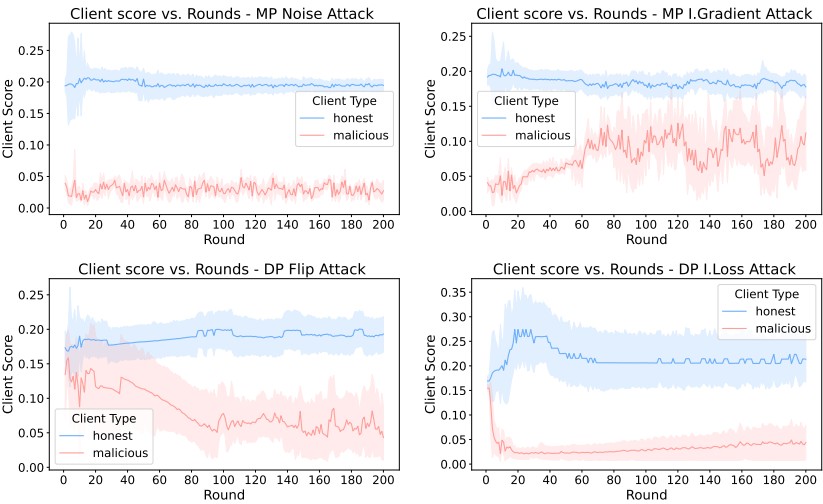

Figure 10: Mean and 95% confidence interval of client scores assigned by our FBSs over 200 rounds on CIFAR-10 across different attacks.

In a similar fashion to Model A, a third model, Model C, may also initially misclassify the point as class 1 but would require a change in features to [2, -1] to alter its prediction to class 0, highlighting different sensitivities and prediction dynamics in response to input variations.

## B.5  Client score visualization

FBPs provide a detailed representation of each client's behaviour, enabling a deeper understanding of the conditions under which malicious clients deviate from the expected behaviour of benign clients. However, in large-scale FL systems, visual inspection may not always be necessary. Instead, statistical metrics derived from these planes can be automatically extracted and analyzed. This is demonstrated in our FBSs, which extract client-specific scores to mitigate or exclude the contributions of malicious clients during aggregation. To demonstrate the scalability and efficacy of automatic detection, we evaluated our approach using the small-CIFAR-10 dataset and recorded client scores throughout the training process using a 5-fold cross-validation scheme. As shown in Figure 10, the extracted scores consistently identify and diminish the influence of malicious clients on the global model, as indicated by the 95% confidence interval, which varies based on the type of attack. Notably, as the model converges (with updates between rounds close to zero), more weight is given to the attacker with an inverted gradient, since its model closely resembles the global model from the previous round (plus a negligible inverted update).

## B.6  Client scoring under unfair server-side validation sets

Our proposed methods rely on a clean validation set on the server to characterize client behaviours. However, in real-world FL scenarios, obtaining a validation set that is entirely fair and representative for each client may not be feasible. This challenge underscores the rationale behind incorporating a multi-plane evaluation in our approach, which assesses client behaviour through two distinct criteria: task performance and counterfactual analysis. Traditional performance metrics in machine learning, such as accuracy, loss, or error, are heavily dependent on the validation set used (i.e., error plane). In contrast, information derived from counterfactuals is more indicative of the learned decision process and provides insight into the training data distribution of each client, even when the validation set is biased (i.e., counterfactual plane). Consequently, an attacker or anomalous client will exhibit distinct behaviours across both planes, particularly showing an unrelated counterfactual distribution compared to other clients. Conversely, an underrepresented client will produce plausible counterfactuals similar to those of other clients, and will only be affected by the unfair validation set in the error plane.

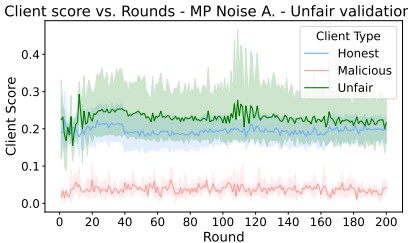

Figure 11: Mean and 95% confidence interval of client scores assigned by our FBSs over 200 rounds on CIFAR-10 with one attacker (MP Noise) and one unfair client (i.e., their data distribution is not in the validation set).

To validate the robustness of our defense mechanism under an unfair validation set, we conducted experiments in which data from one client was excluded from the validation set. We then analyzed the behavioural scores assigned by our method. Preliminary results, shown in Figure 11, indicate that the 95% confidence interval of the scores for the underrepresented client consistently overlaps with those of other honest clients, thereby distinguishing them clearly from malicious clients.

## B.7 Federated Behavioural Shields on IID and non-IID scenarios

In our main study, we focus on the non-IID setting, which is the most prevalent scenario for cross-silo FL in practice [40]. This setting presents considerable challenges due to the substantial differences between updates from honest clients, which can obscure the divergent behaviours of malicious clients [41]. Consequently, we conducted experiments on the Breast Cancer dataset in both IID and non-IID settings using our FBPs for comparative analysis. As depicted in Figure 12, we initially visualised client trajectories on both behavioural planes and within the counterfactual distance space $l$ (defined in Equation 7). The trajectories illustrate the uniform nature of honest client behaviours on both planes, distinctly highlighting the divergent behaviour of the malicious client. Notably, the visualization of the counterfactual distance space, employed by our FBSs, effectively identifies the malicious client. It indicates a high distance of the attacker from all other clients, leading to a low score during the aggregation process. To further validate our approach, we compared the performance of our FBSs under No-attack, Crafted-noise, Inverted-gradient, Label-flipping, and Inverted-loss attacks in both IID and non-IID settings. Table 4 demonstrates that higher accuracy is achieved in the IID setting under almost all conditions compared to the non-IID setting, underscoring the increased complexity of operating in non-IID environments.

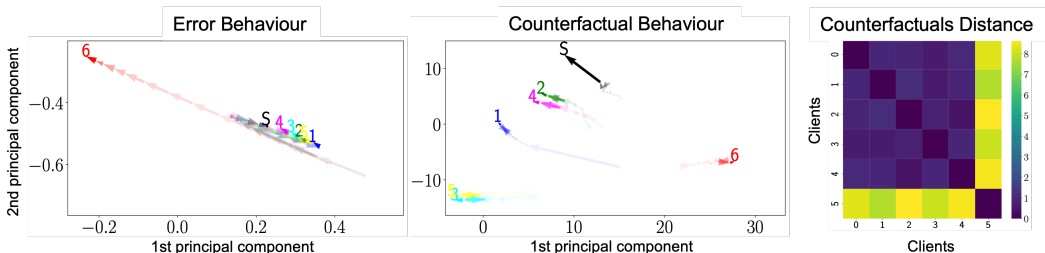

Figure 12: Client trajectories on Counterfactuals and Error Behavioural Planes for Breast Cancer datasets in IID setting under Inverted-loss attacks. The figure highlights the deviation of the malicious client (red - number 6) from honest clients, who tend to cluster together over time.

## B.8 Impact of server validation set size

Considering the crucial role of the validation set size on the server in the creation of behavioural planes, we initially examined its impact on computational time and accuracy through 10 distinct trials. Specifically, we evaluated our FBSs (without moving average) against a malicious client executing an inverted gradient attack on the Diabetes dataset. As depicted in Figure 13, computational

Table 4: Comparison of our FBSs across across five attacks types—No attack, Crafted-noise, Inverted-gradient, Label-flipping, Inverted-loss—on Breast Cancer dataset for both IID and non-IID configurations

| Condition | No attack | MP Noise | MP I.Grad | DP Flip | DP I.Loss | Mean |
|---|---|---|---|---|---|---|
| non-IID | 95.7±1.1 | 98.0±0.8 | 95.3±0.7 | 94.2±0.6 | 95.9±0.9 | 95.8±0.4 |
| IID | 98.2±0.3 | 98.43±0.4 | 98.2±0.2 | 96.4±0.9 | 93.7±1.0 | 97.0±0.4 |

time increases exponentially with the size of the dataset. However, a dataset containing as few as 250 samples is sufficient to adequately represent the data distribution, achieving performance on par with that observed in larger datasets. Additionally, with fewer than 1000 samples, our method demonstrates greater efficiency compared to the Krum algorithm.

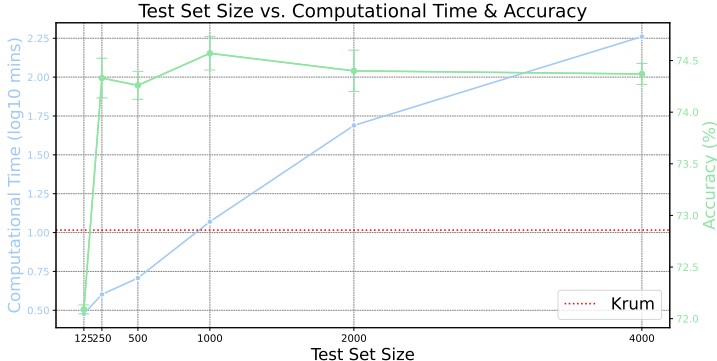

Figure 13: Relationship between test set size, computational time, and accuracy evaluated on Diabetes dataset against inverted gradient attack. As test set size increases (logarithmic x-axis), computational time grows exponentially (log10 scale, red line), while accuracy (blue line) only slightly increases before plateauing.

### B.9 Impact of the window length in the moving average

We analyse the impact of window length on our FBSs and its two variations, one using only predictive performance (error) and the other focusing solely on decision-making processes (counterfactual). The results presented in Figure 14 depict the average accuracy across various conditions—including No attack, Label-flipping, Inverted-loss, Crafted-noise, and Inverted-gradient—for window lengths ranging from 3 to 30 rounds. While no definitive patterns are clearly evident, on average, a longer window length tends to be advantageous for accurately assessing the behavioural scores of each client during the aggregation process. Noteworthy, the optimal accuracy across both datasets was achieved with a window length of 30 rounds. Particularly, a consistent and slight increase in accuracy is observed in the Diabetes dataset when utilising only the counterfactual information.

### B.10 Impact of the local epochs in Federated Behavioural Shields

Local training epochs play a crucial role in FL, particularly in evaluating client behaviours. An increased number of local epochs implies a greater adaptation of the client-model to its local data distribution. For this reason, we assess the performance of our FBSs against data poisoning attacks, including Label-flipping and Inverted-loss, across Breast Cancer, Diabetes, and Synthetic datasets. Metrics are reported as the mean and standard error across ten with distinct parameters' initialization. These attack types were selected because malicious updates are directly influenced by the number of local epochs, similar to updates from honest clients; differently, the behaviour of malicious clients in model poisoning might remain unaffected. As depicted in Figure 15, the impact of local epochs varies across datasets, with each achieving maximum accuracy at different numbers of epochs. Generally, accuracy increases with the number of local epochs until it reaches a peak, after which it begins to

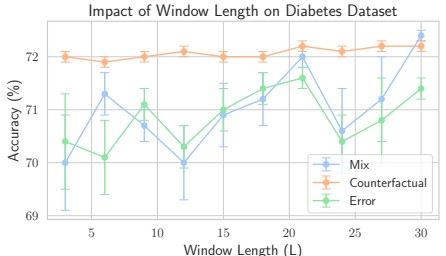
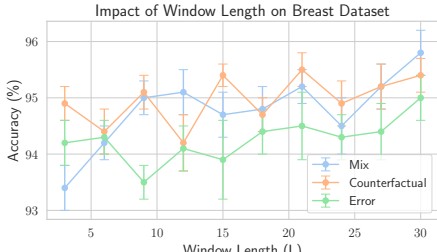

Figure 14: Effect of window length on accuracy using FBSs. This table demonstrates the accuracy changes as a function of window length in the moving average method across the Diabetes and Breast Cancer datasets.

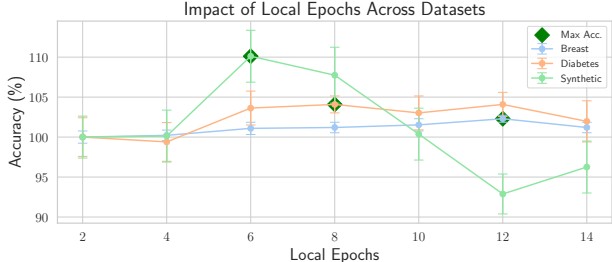

Figure 15: Impact of varying local epochs on accuracy for our FBSs. This table illustrates how changes in the number of local training epochs affect accuracy across the Breast Cancer, Diabetes, and Synthetic datasets.

decline. This trend may be attributed to the fact that initially, increasing the number of local epochs enhances the detectability of anomalies in the behaviour of malicious clients, as their models diverge more quickly from those of honest clients. However, beyond a certain threshold, all client models become highly customd, leading to the potential underweighting of even honest clients' contributions during the aggregation process on the server. This results in the system relying predominantly on a few clients whose models are the most similar to each other.

## B.11 Impact of Attack Intensity on Federated Behavioural Shields

We conducted experiments to evaluate the sensitivity of the proposed method to varying levels of attack intensity. Specifically, we focused on model poisoning attacks by systematically increasing the noise parameter ($\beta$) injected by adversaries into the global model prior to transmission to the server ($noise = \mathcal{N}(0, \beta \cdot \sigma(w^t))$). The evaluation was performed using 5-fold cross-validation on the Breast Cancer and small-MNIST datasets. As shown in Figure 16, increasing the attack intensity results in a noticeable decrease in the performance of the global model when using the standard FedAvg approach. In contrast, our method—both when utilizing all feature planes and when restricted to the counterfactual plane—remains stable and unaffected by the attack intensity. Interestingly, a marginal improvement in accuracy is observed as the attack intensity increases, likely because more aggressive modifications render malicious models more degraded and, therefore, easier to detect.

## B.12 Ablation study

We conducted an ablation study to examine the impact of various components on the efficacy of our algorithm. These components include predictive performance (error), decision-making process (counterfactuals), and the application of a moving average. Table 5 presents the average accuracy and standard error under five experimental conditions: No-attack, Crafted-noise, Inverted-gradient,

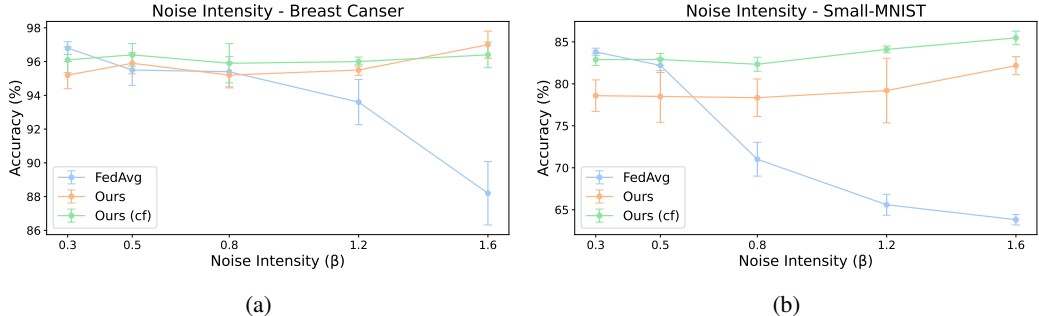

Figure 16: Impact of attack intensity on FBS and traditional FedAvg in (a) the Breast Cancer dataset and (b) the small-MNIST dataset. Notably, unlike FedAvg, FBS maintains stable accuracy as the attack intensity increases.

Label-flipping, and Inverted-loss, comparing different variations of our method against baselines on the Breast Cancer and Diabetes datasets.

Notably, relying solely on predictive performance does not yield a statistically significant benefit compared to the strongest baselines, such as RFA on Breast Cancer and Krum on Diabetes. In contrast, counterfactual information provides deep insights into client behaviour, enabling our method to outperform all baselines in identifying malicious clients. Although the improvement is subtle, the combination of descriptors, on average, provides better results than using counterfactuals alone.

The integration of a moving average notably enhances the performance of our method, improving from $94.5 \pm 0.5$ to $95.4 \pm 0.3$ on the Breast Cancer dataset. Smaller gains are observed on the Diabetes dataset, likely due to two factors: the limited representativeness of the clean validation set on the server for Breast Cancer, which consists of only 89 samples compared to 250 for Diabetes, and the small size of client datasets (77 training samples on average), which may lead to unstable local training. By aggregating behaviour across multiple rounds, the moving average technique helps to stabilise and accurately assess client behaviour scores.

Table 5: Average accuracy (%) ± standard error for various defense strategies under five experimental conditions: No attack, Crafted-noise, Inverted-gradient, Label-flipping, and Inverted-loss. The table specifically compares the performance of our methods with and without the application of a moving average (MA).

| Defense Strategy | Breast | Diabetes |
|---|---|---|
| Krum | $91.9 \pm 0.4$ | $70.6 \pm 0.3$ |
| Median | $92.6 \pm 0.4$ | $67.0 \pm 0.1$ |
| Trimmed-mean | $93.4 \pm 0.4$ | $69.0 \pm 0.1$ |
| RFA | $93.5 \pm 0.5$ | $67.5 \pm 0.1$ |
| Ours (error) w/o MA | $93.7 \pm 0.4$ | $70.5 \pm 0.8$ |
| Ours (cf) w/o MA | $94.2 \pm 0.4$ | $72.0 \pm 0.1$ |
| Ours w/o MA | $94.5 \pm 0.5$ | $72.1 \pm 0.2$ |
| Ours (error) | $95.0 \pm 0.4$ | $71.4 \pm 0.2$ |
| Ours (cf) | $95.3 \pm 0.3$ | $72.2 \pm 0.1$ |
| Ours | $\mathbf{95.8 \pm 0.4}$ | $\mathbf{72.4 \pm 0.1}$ |

## B.13 Comprehensive tabular analysis of defense mechanisms

In this section, we provide a detailed numerical breakdown of the results depicted in Figure 4. This quantitative analysis aims to supplement the visual data presented, offering precise values and statistical insights that underpin the observations and conclusions discussed throughout the paper. Tables 6 for Breast Cancer, 7 for Diabetes, 8 for small-MNIST, and 9 for small-CIFAR-10 present a comprehensive comparison of our FBSs and its streamlined versions using only the CBP or the EBP. These are evaluated against traditional defenses such as Krum, Median, Trimmed-mean, RFA across

five scenarios: No attack, Crafted-noise, Inverted-gradient, Label-flipping, and Inverted-loss. This comparison elucidates the effectiveness of our approach under a range of conditions, both adversarial and non-adversarial.

Table 6: Comparison of FBSs and its simpler version with only counterfactuals (cf) or predictive-performance (error) versus Krum, Median, and Trimmed-mean defenses across five attacks types—No attack, Crafted-noise, Inverted-gradient, Label-flipping, Inverted-loss—on Breast Cancer dataset.

| | | Model Poisoning | | Data Poisoning | | |
| --- | --- | --- | --- | --- | --- | --- |
| | No-Attack | Crafted-Noise | Inv. Grad. | Label-Flip | Inv. Loss | Mean |
| Krum | 90.9±0.6 | 92.3±0.8 | 92.7±1.5 | 91.6±0.4 | 91.8±0.6 | 91.9±0.4 |
| Median | 92.7±0.7 | 93.6±0.7 | 93.1±1.2 | 92.0±0.8 | 91.6±1.1 | 92.6±0.4 |
| Trim | 93.2±0.5 | 96.1±0.7 | 95.5±0.4 | 91.8±1.1 | 90.4±0.8 | 93.4±0.3 |
| RFA | 94.3±1.5 | 94.1±0.9 | 93.6±1.1 | 93.6±0.9 | 91.8±1.0 | 93.5±0.5 |
| Ours (error) | 96.6±0.8 | 97.7±1.2 | 95.0±0.8 | 91.4±1.0 | 94.1±0.8 | 95.0±0.4 |
| Ours (cf) | 97.5±0.5 | 96.8±0.6 | 96.6±0.3 | 93.2±0.9 | 92.3±0.6 | 95.3±0.3 |
| Ours | 95.7±1.1 | 98.0±0.8 | 95.3±0.7 | 94.2±0.6 | 95.9±0.9 | 95.8±0.4 |
| Predictor | 96.6± 0.4 | *N/A* | *N/A* | *N/A* | *N/A* | *N/A* |

Table 7: Comparison of FBSs and its simpler version with only counterfactuals (cf) or predictive-performance (error) versus Krum, Median, and Trimmed-mean defenses across five attacks types—No attack, Crafted-noise, Inverted-gradient, Label-flipping, Inverted-loss—on Diabetes dataset.

| | | Model Poisoning | | Data Poisoning | | |
| --- | --- | --- | --- | --- | --- | --- |
| | No-Attack | Crafted-Noise | Inv. Grad. | Label-Flip | Inv. Loss | Mean |
| Krum | 70.5±1.2 | 70.4±0.2 | 71.0±0.2 | 70.5±0.2 | 70.7±0.1 | 70.6±0.3 |
| Median | 69.7±0.2 | 68.5±0.3 | 69.2±0.1 | 64.1±0.1 | 65.1±0.2 | 67.3±0.1 |
| Trim | 67.9±0.1 | 69.6±0.4 | 71.7±0.2 | 65.1±0.1 | 70.5±0.2 | 69.0±0.1 |
| RFA | 70.5±0.2 | 69.7±0.2 | 69.8±0.2 | 65.1±0.2 | 62.2±0.1 | 67.5±0.1 |
| Ours (error) | 71.4±0.6 | 72.0±0.5 | 72.2±0.4 | 72.3±0.2 | 69.3±0.8 | 71.4±0.2 |
| Ours (cf) | 72.7±0.2 | 72.3±0.2 | 72.4±0.0 | 71.8±0.2 | 71.6±0.2 | 72.2±0.1 |
| Ours | 72.5±0.3 | 73.1±0.4 | 72.3±0.0 | 72.0±0.4 | 71.9±0.3 | 72.4±0.1 |
| Predictor | 73.5±0.2 | *N/A* | *N/A* | *N/A* | *N/A* | *N/A* |

Table 8: Comparison of FBSs and its simpler version with only counterfactuals (cf) or predictive-performance (error) versus Krum, Median, and Trimmed-mean defenses across five attacks types—No attack, Crafted-noise, Inverted-gradient, Label-flipping, Inverted-loss—on small-MNIST dataset.

| | | Model Poisoning | | Data Poisoning | | |
| --- | --- | --- | --- | --- | --- | --- |
| | No-Attack | Crafted-Noise | Inv. Grad. | Label-Flip | Inv. Loss | Mean |
| Krum | 56.3±2.4 | 60.3±1.3 | 62.0±1.5 | 59.3±2.5 | 55.9±3.0 | 58.8±1.0 |
| Median | 68.8±1.5 | 69.5±2.0 | 63.7±1.6 | 64.2±1.7 | 64.5±1.0 | 66.1±0.7 |
| Trim | 68.7±2.4 | 78.9±2.3 | 77.7±2.0 | 63.9±1.9 | 70.1±3.0 | 71.9±1.1 |
| RFA | 73.8±2.9 | 72.2±1.8 | 74.4±1.4 | 66.6±5.3 | 72.0±2.1 | 71.8±1.4 |
| Ours (error) | 80.2±1.5 | 71.0±4.2 | 71.3±0.3 | 76.0±2.2 | 72.9±4.7 | 74.3±1.4 |
| Ours (cf) | 84.2±0.2 | 84.1±0.4 | 74.5±1.7 | 77.8±1.7 | 80.1±0.6 | 80.1±0.5 |
| Ours | 81.0±2.0 | 79.2±3.8 | 74.6±0.6 | 74.6±3.7 | 77.4±2.5 | 77.4±1.2 |
| Predictor | 79.6±0.8 | *N/A* | *N/A* | *N/A* | *N/A* | *N/A* |

Table 9: Comparison of FBSs and its simpler version with only counterfactuals (cf) or predictive-performance (error) versus Krum, Median, Trimmed-mean, and RFA defenses across five attack types—No attack, Crafted-noise, Inverted-gradient, Label-flipping, Inverted-loss—on small-CIFAR-10 dataset.

| | | Model Poisoning | | Data Poisoning | | |
| | No-Attack | Crafted-Noise | Inv. Grad. | Label-Flip | Inv. Loss | Mean |
|---|---|---|---|---|---|---|
| Krum | 90.6±2.0 | 90.7±1.3 | 92.1±1.1 | 93.0±1.3 | 85.3±2.4 | 90.3±0.3 |
| Median | 96.5±0.4 | 97.0±0.4 | 96.1±0.2 | 96.0±0.7 | 80.6±5.3 | 93.2±0.5 |
| Trim | 96.5±0.3 | 96.9±0.3 | 97.4±0.3 | 96.2±0.6 | 86.3±3.4 | 94.7±0.3 |
| RFA | 96.3±0.4 | 97.2±0.3 | 97.0±0.3 | 96.3±0.8 | 83.8±3.6 | 94.1±0.3 |
| Ours (error) | 97.4±0.2 | 97.1±0.2 | 96.1±0.4 | 97.3±0.5 | 89.8±1.2 | 95.5±0.1 |
| Ours (cf) | 98.2±0.2 | 97.4±0.3 | 97.6±0.1 | 97.6±0.2 | 91.8±0.8 | 96.5±0.1 |
| Ours | 97.4±0.3 | 97.9±0.4 | 97.3±0.3 | 97.2±0.4 | 91.7±1.0 | 96.3±0.1 |
| Predictor | 97.5±0.3 | *N/A* | *N/A* | *N/A* | *N/A* | *N/A* |

