# OpenReview forum: "Federated Behavioural Planes: Explaining the Evolution of Client Behaviour in Federated Learning"
_NeurIPS.cc/2024/Conference — NeurIPS 2024 poster_

### Official Review · Reviewer_aREF · 2024-07-01

**Soundness:** 3
**Presentation:** 3
**Contribution:** 2
**Rating:** 5
**Confidence:** 2

**Summary:**

The paper introduces Federated Behavioural Planes (FBPs), a method designed to track FL clients' behavior by means of examining the their representations in two behavioural planes with the aid of a server-owned dataset. The Error Behavior Plane (EBP) and Counterfactual Behavioural Plane (CBP) correspond to two 2-D space where each client model's prediction errors and counterfactural distribution are examined. Combining these two representations a new aggregation rule is proposed to fend off malicious clients.

**Strengths:**

- The paper is overall well written. The idea presented is clear and easy to follow.
- The method of characterizing clients' behavior is kind of novel.

**Weaknesses:**

- Creating the two plains require the plaintext of all local models, which raises major privacy concerns.
- From Eqs. 6 and 7, I suppose generating the counterfacturals must be very costly.
- Insufficient experimental evidence to support the claim. For example, fig. 3 does not show distinct client behavior clearly. I also expect to see more results such as the behavioural scores of different clients and how sensitive the FBP is to the intensity of attacks.

**Questions:**

- Results in Table 1 make me confused whether the whole method is still run in a cannonical FL setting.
- What is the difference between proximity and sparsity?
- How is counterfactural generator trained and why does it impact the task accuracy?
- What purpose does Eq. 4 serve?

**Limitations:**

- A framework overview is missing, which makes it hard to understand how the planes are created throughout the FL iterations.

---

> ### Author Rebuttal · Authors · 2024-08-06
>
> _Privacy concerns of transmitting the plaintext of all local models_
>
> __Our method can integrate Local Differential Privacy (LDP) or Homomorphic Encryption (HE) to enhance privacy, ensuring that sensitive information remains protected while maintaining robustness.__
> Our framework, like traditional FedAvg, is susceptible to privacy leakage if plaintext local model parameters are exposed. However, it can incorporate LDP, where clients add noise to the model before transmission, maintaining DP while creating behavioral planes. Although this process may result in some performance loss, it is comparable to other robust aggregation methods that calculate client similarities using differentially private model parameters.
> To further enhance privacy, our framework can integrate HE, which masks client models while allowing for inference on the validation set. Although this increases computational time during the aggregation, it ensures that sensitive information remains completely concealed from the server, thereby addressing privacy concerns effectively. We added a sentence in L348: "Since our method allows the integration of privacy-enhancing techniques such as LDP and HE, a promising future direction would be to analyze their impact on performance and efficiency".
>
> _Insufficient experimental evidence regarding the distinction of client behaviors through Behavioral Planes (BPs). Expected more results, e.g. the client behavioral scores_
>
> __We have strengthened the experimental evidence by analyzing behavioral scores extracted from BPs, demonstrating that our method accurately identifies and mitigates malicious clients across all attacks.__
> As suggested, we analyze the behavioral scores of honest and malicious clients across all attacks (Fig. B), which are calculated over the BPs. The 5-fold experiments were conducted on the CIFAR10 (most complex dataset), and show the mean and 95% confidence intervals during the training rounds. Statistically, our method consistently identifies malicious clients, effectively excluding or reducing their weights during aggregation. These results further demonstrate that BPs accurately describe malicious clients during the training process.
>
> _How sensitive the proposed method is to the intensity of the attacks_
>
> __Our method remains stable and unaffected by increasing attack intensity, demonstrating robustness against model poisoning attacks.__
> We analyzed the sensitivity of our method to attack intensity by increasing the amount of noise $\beta$ that attackers add to the global model before sending it to the server [L677]. We conducted 5-fold experiments on both the Breast Cancer and MNIST datasets. As shown in Fig. E, increasing attack intensity decreases FedAvg’s performance, while our methods (using all planes and only the counterfactual plane) remain stable and unaffected by the attack intensity. Interestingly, our methods’ accuracy slightly increased as attack intensity rose, likely because the malicious models became more degraded and, therefore, less stealthy.
>
> _A framework overview is missing, which makes it hard to understand how the planes are created throughout the FL iterations_
>
> We thank the reviewer for this observation. To address this limitation, __we have provided a detailed algorithm (Algorithm 1) for our method to create the behavioral planes and implement our robust aggregation (FBSs)__. While Algorithm 1 covers the server operations, we will also include the algorithm for the client operations in the Appendix A.5 of the paper, which reflects the traditional FL algorithm.
>
> _Results in Table 1 make me confused whether the whole method is still run in a cannonical FL setting_
>
> __Yes, our method operates within a canonical FL setting, with changes to server-side aggregation and the introduction of the counterfactual (CF) generator. The CF generator is trained jointly with the local client and does not alter the performance of the predictor alone (Table 1)__. For clarity, we updated L230-231: "We compared model accuracy across four settings: two centralized learning (CL) and two FL, using three different datasets under non-IID conditions. For FL experiments, we used the traditional FedAvg approach with two variations: predictor-only and predictor with CF generator."
>
> _What is the difference between proximity and sparsity?_
>
> These are standard metrics to quantitatively evaluate counterfactuals (CFs) [19, 35 in the paper]. __Proximity measures how much our CFs are closer to our data distribution (distance between the CF and the closest data point in the training set with the same label), while sparsity measures how many features were changed between the initial sample and the CF.__ For clarity, we updated L208-209: “proximity (↓) [35], assessing the realism of CFs by their closeness to the training data (distance between the CF and the closest data point in the training set with the same label); and sparsity (↓) [19], which quantifies the changes made to the input to generate the CFs (number of features changed between the initial sample and the CF).”
>
> _What purpose does Eq. 4 serve?_
>
> __Eq. 4 provides a theoretical foundation for understanding FL dynamics, describing how client behaviors evolve during training and are influenced by intermediate steps.__
> As mentioned by both Rev-eMJr and ULpn, our primary goal is to provide a general theoretical foundation for understanding FL dynamics, specifically how client behaviors evolve during training and how they are influenced by intermediate steps (e.g., local training and aggregation). Eq. 4 describes the dynamics of client behaviors in the FL system. Although we did not analytically solve it, we experimentally observe its solutions through client behaviors using our planes. For clarity, we modify L97-98: "These dynamics can be encapsulated in the following differential equation, which describes how client behaviors evolve during training and are influenced by internal forces within the FL system".

---

> ### Author Response · Authors · 2024-08-12
>
> Thank you for your valuable feedback. Please let us know if you have any further questions or if there are any points that need additional clarification. We would be grateful if you could consider updating your review in light of our responses.

---

### Official Review · Reviewer_ULpn · 2024-07-02

**Soundness:** 3
**Presentation:** 2
**Contribution:** 2
**Rating:** 6
**Confidence:** 3

**Summary:**

This paper introduces a novel method called Federated Behavioural Planes (FBPs) for analyzing, visualizing, and explaining the dynamics of Federated Learning (FL) systems. FBPs are consist of Error Behavioural Plane (EBP), reflecting the model’s predictive performance, and Counterfactual Behavioural Plane (CBP), reflecting the decision-making processes. Using insights from FBPs, the paper also proposes a new robust aggregation technique called Federated Behavioural Shields (FBS) to enhance security in FL systems.

**Strengths:**

+ Novel approach: The Federated Behavioural Planes (FBPs) introduce a new way to analyze and visualize client behaviors in Federated Learning, addressing a gap in existing literature.
+ Theoretical foundation: The authors provide a theoretical framework for understanding FL dynamics, grounding their practical approach in solid mathematical concepts.
+ Explanatory power: FBPs allow for visual identification of client clusters and trajectories, enhancing interpretability of FL systems.

**Weaknesses:**

- The paper heavily relies on visual representations (FBPs) to explain client behavior, which makes it infeasible to scale to large-scale FL systems where the convergence might take more rounds and client selection takes place in each round. These factors will largely make the trajectory on the planes hard to keep track of.

&nbsp;

- The computational overhead is too large.
  + In each training round, the counterfactuals are required to be computed for the locally updated model from every client. And the differences between the counterfactuals from each pair of clients are also computed. For cross-silo FL settings, this is already a large computational overhead. For cross-device settings, I do not think this process is affordable especially for high dimensional data like image data.
  + Besides, since the paper focuses primarily on detecting anomalies and enhancing security from attackers, the main battlefield of this method should be more on the cross-device setting where the number of clients is large. To this end, the prohibitive computational overhead mitigates the motivation of this work.

&nbsp;

- The method completely relies on a validation set on the server, which may be hard to obtain without introducing any privacy concerns. Even a validation set is available to use, under what circumstances will the validation set be equally fair to all clients so that it is not far from any client’s local data distribution. And if the plane shows anomalies for some client, how to distinguish the reasons from having an unfair validation set and client is the anomaly/attacker?

&nbsp;

- The datasets and the federated settings in the experiments are too simple. And the compared robust aggregation methods are not so recent.

**Questions:**

- Is the counterfactual generator learned on the server? Why would a generator whose goal is for better interpretability affect the performance of predictor?
- How is the local CL’s accuracy computed, and why is it so much lower than FL in Table 1?

**Limitations:**

Most limitations of the proposed method (validation set, computational overhead) have been addressed. Please refer to Weaknesses for other limitations of this work. There is no potential societal impact.

---

> ### Author Rebuttal · Authors · 2024-08-07
>
> _The paper heavily relies on visual representations to explain client behaviors, infeasible for large FL systems._
>
> __Our method is scalable and automatically extracts and analyzes statistics from behavioral planes (BPs) to identify and mitigate malicious clients without relying on visual inspection. Visual inspection aids in understanding the characteristics under which malicious clients behave differently, but it is not necessary.__ Our defense extracts client-specific scores from BPs in large FL systems to reduce the impact of malicious clients. To further demonstrate scalability and automatic identification of malicious behaviors, we recorded client scores during training using CIFAR10 (Fig. B). These scores can identify malicious clients and reduce their contribution to the global model. Interestingly, as the model converges, more weight is given to the attacker with an inverted gradient, since its model closely resembles the global model from the previous round (plus a negligible inverted update). Visual inspection can still be useful for ML engineers to understand why certain clients are removed or contribute less, by visualizing on multiple planes where they differ from ‘honest’ clients (see Fig. 3). The experiments and discussion will be added to Appendix B.
>
> _Client selection at each round will affect trajectory on the planes._
>
> __Our method ensures that client selection does not impact the effectiveness of our defense (FBSs), as aggregation relies on both current and available historical behavior. Visualizations can track client trajectories based on selected rounds, maintaining a clear representation of client behavior.__ If a client is not selected in a round, its trajectory will not consider that particular round, but trajectories can still be created based on selected rounds. While the current focus is on evaluating client behavior over time, it is also possible to establish trajectories only for selected clients relative to the server’s position, ensuring accurate visualization not affected by client selection. Finally, although client selection influences the visual representation, it has no effect on FBSs, as aggregation depends on both current and historical client behavior (Moving Average in Sec. 3.5).
>
> _The method relies on a server validation set, which may be difficult to obtain without privacy concerns._
>
> __Several solutions can be integrated with our method to create a synthetic validation set on the server without transmitting sensitive information from clients.__ A clean validation set is a common assumption of existing methods [51, 45, 31 cited in paper]. To address this, some proposed solutions include optimizing for a clean validation set [15] and generating representative inputs through dimensionality reduction and stratified sampling [44].
> As discussed in Sec. 6.2, we plan to:
> - Apply an autoencoder or Bayesian probability conditioned on class labels to generate a synthetic dataset based on client data distributions
> - Explore synthetic generation as in [44]
>
> If the server has a validation set, our method weights each client to maximize global model performance on that test set, regardless of whether the model weights come from honest or malicious clients
>
> _When is the validation set fair to all clients? How to distinguish between an unfair validation set and an anomaly/attacker?_
>
> This is a great question, and that’s exactly why we proposed the multi-plane evaluation. Unlike traditional ML metrics such as accuracy or error, __our defense method is not strongly influenced by the fairness of the validation set, as counterfactuals (CFs) are more closely related to the learned client decision process and data distribution than the evaluation set.__ Anomalous clients show unrelated CF distributions and high error, while underrepresented clients generate plausible CFs but are affected by the unfair validation set only in the error plane. To demonstrate effectiveness under an unfair validation set, we removed one client’s data from the validation set and recorded behavioral scores. Preliminary results (Fig. C) show that the 95% confidence interval of the unfair client’s score overlaps with those of other honest clients, distinguishing them from malicious clients. We introduce in L340: Considering the importance of fairness among clients and the promising results shown in Appendix B, the exploration of the impact of unfair validation sets and the development of other validation-independent descriptors, such as CFs, represent promising areas for future research.
>
> _Datasets are simple and baselines are not so recent_
>
> __We have enhanced our experimental setup by introducing a more challenging dataset, CIFAR-10, with a 10-class classification task, and a recent robust aggregation baseline, RFA [Pillutla et al., 2022], into all our experiments to further demonstrate the robustness and effectiveness of our method.__ Even under these new conditions (Fig. A), our method outperforms or matches other baselines across all four datasets and all five attack conditions, except for Inv. gradient attacks in MNIST, where Trimmed mean performs better (1 out of 20 cases).
>
> _Is the counterfactual generator learned on the server?_
>
> __No, the generator is simultaneously trained in an end-to-end fashion on the client-side along with the predictor__ [L139,L231]. Please refer to the common answers for model analysis.
>
> _How is the local CL’s accuracy computed, and why is it so much lower than FL in Table 1?_
>
> __In Local CL, we train a separate model for each client’s private data__, ensuring privacy by not sharing data across clients. In non-IID settings, training on single-client data reduces performance due to the lack of diverse data, explaining the lower accuracy of Local CL compared to FL, which uses data from all clients, in Table 1. We added in L235: For Local CL, we report the average accuracy of all models trained individually by each client and evaluated on a common test set

---

> > ### Comment · Reviewer_ULpn · 2024-08-12
> > **Thanks for the rebuttal**
> >
> > Thanks the rebuttal. Based on the author's rebuttal, I believe I have a better understanding of the content of the paper. Most of my concerns have been addressed and I have increased my rating.

---

> ### Author Response · Authors · 2024-08-12
>
> Thank you for your valuable feedback. Please let us know if you have any further questions or if there are any points that need additional clarification. We would be grateful if you could consider updating your review in light of our responses.

---

### Official Review · Reviewer_eMJr · 2024-07-13

**Soundness:** 3
**Presentation:** 3
**Contribution:** 3
**Rating:** 7
**Confidence:** 4

**Summary:**

This paper introduces a novel method called Federated Behavioural Planes (FBPs) to analyze, visualize, and explain the dynamics of client behavior in Federated Learning (FL) systems. The primary contributions of the paper are as follows:

* 1\. Introduction of Federated Behavioural Planes (FBPs): FBPs consist of two planes:
     * 1.1\. Error Behavioural Plane (EBP): This plane analyzes the predictive performance of client models by visualizing the errors they produce.

     * 1.2\. Counterfactual Behavioural Plane (CBP): This plane examines the decision-making processes of client models through counterfactual explanations, highlighting how decision boundaries are formed.
* 2\. Visualization and Analysis: FBPs provide informative trajectories that describe the evolving states of clients, enabling the identification of clusters of clients with similar behaviors. This helps in understanding both beneficial and detrimental behaviors in FL systems.

* 3\. Federated Behavioural Shields (FBS): Based on the patterns identified by FBPs, the authors propose a robust aggregation technique named Federated Behavioural Shields. This technique enhances security by detecting malicious or noisy client models and surpasses the efficacy of existing state-of-the-art FL defense mechanisms.

* 4\. Experimental Validation: The paper demonstrates through experiments that FBPs can effectively track client behavior, identify client clusters, and improve the security and performance of FL systems. The proposed FBS method outperforms other robust aggregation methods in defending against various types of attacks.

Overall, the paper offers a comprehensive approach to enhance understanding, trust, and control over federated learning systems by introducing a novel method to analyze and secure client behaviors.

**Strengths:**

The paper "Federated Behavioural Planes: Explaining the Evolution of Client Behaviour in Federated Learning" demonstrates several strengths across different dimensions:

* 1\. Originality: The paper introduces a novel approach, Federated Behavioural Planes (FBPs), to analyze and visualize client behavior dynamics in Federated Learning systems. This method offers a unique perspective on understanding client behavior evolution in FL, which is a relatively unexplored area in the existing literature. The combination of predictive performance analysis and decision-making process evaluation through FBPs showcases originality in addressing the challenges of client behavior in FL systems.
* 2\. Quality: The paper maintains a high standard of quality in terms of methodology, experimental design, and theoretical framework. The introduction of FBPs as a tool to explain the dynamics of FL systems reflects a well-thought-out approach to addressing the evolving behavior of clients in federated learning environments. The robust aggregation mechanism proposed, Federated Behavioural Shields, demonstrates a quality solution to enhance security in FL systems.
* 3\. Clarity: The paper is well-written and structured, making it easy for readers to follow the concepts presented. The clarity in explaining the Federated Behavioural Planes framework, the experimental results, and the implications of the proposed method enhances the overall understanding of the research. The use of figures and explanations aids in visualizing complex concepts related to client behavior in FL systems.
* 4\. Significance: The paper's contribution to the research area of Federated Learning is significant. By introducing FBPs and Federated Behavioural Shields, the paper addresses a key challenge in FL systems - understanding and controlling client behavior. The insights provided by FBPs and the improved security offered by Federated Behavioural Shields have the potential to enhance the reliability and control over FL systems, making a valuable contribution to the field.

Overall, the paper makes a substantial contribution to the field of federated learning. Its originality lies in the novel problem formulation and creative combination of existing ideas. The quality of the research is demonstrated through rigorous methodology and comprehensive experiments. The clarity of the presentation ensures that the contributions are accessible to a broad audience. The significance of the work is underscored by its potential impact on improving the security and efficiency of federated learning systems.

**Weaknesses:**

**Complexity of Methods:**
 - **Computational Overhead**: The concurrent training of counterfactual generators with the main predictive models introduces significant computational overhead. This could be particularly burdensome in real-world federated learning settings where resources are limited. The paper could benefit from a more detailed analysis of the computational costs and potential optimization strategies to mitigate this overhead.
    - **Actionable Insight**: Consider providing a detailed comparison of the computational requirements of the proposed method with baseline methods. Explore possible optimizations or approximations that could reduce the overhead without significantly compromising the performance.

**Real-World Applicability:**
- **Scalability Concerns**: While the experiments are comprehensive, they are conducted on relatively small datasets and a limited number of clients. This raises concerns about the scalability of the proposed methods to larger, real-world federated learning scenarios with many clients and more complex data distributions.
    - **Actionable Insight**: Include a discussion on the scalability of FBPs and FBS. Consider performing a scalability analysis, even if only theoretical, to predict the performance and feasibility of the methods in larger settings. Additionally, simulations or theoretical models could provide insights into expected behavior in large-scale deployments.

**Generalization Across Different Models:**
- **Model-Specific Limitations**: The proposed method may be tailored to specific types of models (e.g., neural networks) and may not generalize well to other types of models used in federated learning (e.g., decision trees, support vector machines).
    - **Actionable Insight**: Discuss the applicability of FBPs and FBS to different types of models. Providing a broader range of experiments that include different model architectures could strengthen the paper. If certain models are not compatible, explain the limitations and potential modifications required for broader applicability.

**Evaluation Metrics:**
- **Limited Evaluation Metrics**: The evaluation primarily focuses on standard metrics like accuracy and robustness against attacks. While these are important, they might not capture all aspects of the system's performance, such as the impact on communication efficiency, latency, and energy consumption.
    - **Actionable Insight**: Introduce additional evaluation metrics that capture the holistic performance of the system, including communication overhead, latency, and energy consumption. This would provide a more comprehensive assessment of the practicality of the proposed methods in real-world applications.

**Interpretability and Usability:**
- **Interpretability for Non-Experts:** The paper, while clear in its technical explanations, may still be challenging for practitioners who are not experts in federated learning or explainable AI. Enhancing the interpretability and usability of the methods for a broader audience could be beneficial.
    - **Actionable Insight:** Provide more intuitive explanations and visualizations of the key concepts and methods. Including case studies or practical examples demonstrating the application of FBPs and FBS in real-world scenarios could make the methods more accessible and easier to understand for non-experts.

**Real-World Validation:**
- **Lack of Real-World Validation**: The experiments are conducted in controlled settings, which may not fully represent the challenges and variability encountered in real-world federated learning deployments.

    - **Actionable Insight**: Discuss potential real-world applications and the expected challenges. If possible, provide preliminary results or insights from deploying the methods in a real-world scenario. Alternatively, outline a detailed plan for future real-world validation studies.

**Conclusion:**
While the paper presents significant contributions to federated learning, addressing these weaknesses could enhance its impact and practicality. By focusing on computational efficiency, scalability, broader model applicability, comprehensive evaluation metrics, interpretability, and real-world validation, the work can move closer to its stated goals and become more robust and applicable to a wider range of scenarios.

**Questions:**

**Questions for the Authors:**
* 1\. **Computational Overhead:**
    - Question: How significant is the computational overhead introduced by the concurrent training of counterfactual generators with the main predictive models?
    - Suggestion: Provide a detailed comparison of the computational requirements of your method with baseline methods. Include any potential optimization strategies to mitigate this overhead.

* 2\. **Scalability:**
    - Question: How does the proposed method scale with a larger number of clients and more complex datasets?
    - Suggestion: Include a scalability analysis or discussion that addresses the performance and feasibility of FBPs and FBS in larger, real-world federated learning scenarios. Simulations or theoretical models predicting the system's behavior in large-scale deployments would be helpful.

* 3\. **Applicability to Different Models:**
    - Question: Is the proposed method applicable to different types of models beyond neural networks, such as decision trees or support vector machines?
    - Suggestion: Discuss the generalizability of FBPs and FBS to various model types. If there are limitations, explain the necessary modifications to apply the method to other model architectures.

* 4\. **Evaluation Metrics:**

    - Question: Have you considered additional evaluation metrics that capture communication efficiency, latency, and energy consumption in your analysis?
    - Suggestion: Introduce and discuss additional metrics to provide a comprehensive assessment of the system’s performance in practical applications. This would strengthen the evaluation of the proposed methods.

* 5\. **Interpretability and Usability:**

    - Question: How can the methods be made more interpretable and usable for practitioners who are not experts in federated learning or explainable AI?
    - Suggestion: Provide more intuitive explanations and visualizations. Including case studies or practical examples of FBPs and FBS in real-world scenarios could enhance understanding and applicability for a broader audience.

* 6\. **Real-World Validation:**

    - Question: Have you conducted any preliminary real-world validations of the proposed methods? If not, what are the plans for such validations?
    - Suggestion: Discuss any real-world applications or preliminary results. Outline a detailed plan for future real-world validation studies to demonstrate the practical applicability and effectiveness of your methods.

* 7\. **Comparison with State-of-the-Art:**

    - Question: How does the proposed Federated Behavioural Shields compare with other state-of-the-art defense mechanisms under different attack scenarios?
    - Suggestion: Provide a detailed comparative analysis with other robust aggregation methods under various attack types. Highlight the strengths and potential weaknesses of your approach in comparison to existing techniques.

* 8\. **Visualizations and Trajectories:**

    - Question: Can you provide more detailed examples or visualizations of client trajectories in the Error Behavioural Plane and Counterfactual Behavioural Plane?
    - Suggestion: Include more visual examples and explanations of client trajectories to illustrate how FBPs can be used to identify different client behaviors and detect malicious clients.

* 9\. **Impact of Non-IID Data:**

    - Question: How does the method handle non-IID (non-independent and identically distributed) data across clients, and how robust is it to such scenarios?
    - Suggestion: Discuss the impact of non-IID data distributions on the performance of FBPs and FBS. Provide experimental results or theoretical analysis demonstrating the method’s robustness to non-IID data.

* 10\. **Future Directions:**

    - Question: What are the future directions or potential extensions of your work?
    - Suggestion: Outline possible future research directions or extensions of FBPs and FBS. This could include integrating additional behavioral planes, optimizing computational efficiency, or exploring new applications of the methods.

**Conclusion:**
Addressing these questions and suggestions can provide clarity on the strengths and limitations of the proposed methods, enhance the understanding of their practical applicability, and offer insights into potential improvements. Engaging with these points during the rebuttal phase can lead to a productive discussion and potentially strengthen the overall contribution of the paper.

**Limitations:**

**Assessment of Limitations and Potential Negative Societal Impact:**

Based on the provided content and the NeurIPS checklist guidelines on limitations and broader societal impacts, here is an assessment of how well the authors have addressed these aspects and suggestions for improvement:

**Addressing Limitations:**

- **Identified Limitations:** The paper acknowledges some limitations, such as the computational overhead introduced by the concurrent training of counterfactual generators and the potential scalability issues in larger federated learning deployments.

- **Suggestions for Improvement:**
    - **Computational Efficiency:** Provide a more detailed discussion on the computational requirements and potential optimization strategies. This could include specific techniques to reduce overhead, such as model pruning, quantization, or distributed optimization methods.

    - **Scalability Analysis:** Include more comprehensive scalability experiments or simulations that predict the method's performance in larger, real-world settings. Discuss how the method can be adapted or optimized for large-scale deployments.

**Potential Negative Societal Impact:**
- **Security and Privacy:** The primary focus of the paper is on enhancing security and privacy in federated learning, which is a positive societal impact. However, potential negative impacts, such as the misuse of federated learning systems or unintended biases in the models, should be considered.

- **Suggestions for Improvement:**
    - **Bias and Fairness:** Discuss the potential for unintended biases in federated learning models and how the proposed methods could mitigate or exacerbate these biases. Provide suggestions for ensuring fairness in federated learning deployments.
    - **Misuse of Technology:** Address the potential for misuse of federated learning systems, such as using the technology for surveillance or other harmful purposes. Discuss safeguards and ethical considerations to prevent misuse.

**Constructive Suggestions for Improvement:**
* 1\. **Detailed Discussion on Limitations:**

    - Expand the discussion on identified limitations, providing more details on computational efficiency and scalability. Include theoretical analysis or empirical evidence supporting the claims and potential solutions.
* 2\. **Bias and Fairness:**

    - Add a section discussing the potential for biases in federated learning models. Explain how the proposed methods could impact fairness and provide guidelines or best practices for ensuring equitable outcomes.
* 3\. **Ethical Considerations and Misuse:**

    - Address potential misuse of federated learning technology. Discuss ethical considerations and propose safeguards to prevent the harmful application of the technology. Highlight the importance of transparency and accountability in federated learning deployments.
* 4\. **Societal Impact Statement:**

    - Include a comprehensive societal impact statement that covers both positive and negative aspects. This should highlight the benefits of enhanced security and privacy, as well as address the potential risks and ethical concerns.

**Conclusion:**
While the paper makes significant contributions to enhancing security and privacy in federated learning, it could benefit from a more thorough discussion of limitations and potential negative societal impacts. By addressing these areas, the authors can provide a more balanced view of their work and ensure that it is applied responsibly and ethically.

---

> ### Author Rebuttal · Authors · 2024-08-06
>
> _Computational Overhead and Evaluation Metrics, the paper could benefit from a detailed analysis of the computational costs._
>
> As suggested, __we performed a detailed analysis of the computational overhead of our framework, examining all its components__: Local Computation, Communication Overhead, and Server-side Computation. Our evaluation metrics include GFLOPs for local inference, megabytes for communication, time complexity for the aggregation process, and the duration of one training round. We also conducted a comprehensive comparison with other methods. Please refer to the common comment (@Rev-eMJr,ULpn,aREF) for detailed results and discussions.
>
>
> _Scalability concerns. “While the experiments are comprehensive, they are conducted on relatively small datasets” Question: “How does the proposed method scale with a larger number of clients and more complex datasets?”_
>
> __We enhance the robustness of our method’s validation by: 1) Introducing a more complex dataset, CIFAR-10. 2) Adding a new baseline, [K. Pillutla et al., RFA, 2022], for all experiments. 3) Experimentally demonstrating that our methods scale effectively with a larger number of clients.__
> We introduced an experiment on CIFAR10, which is a larger dataset and a recent FL benchmark [K. Pillutla et al., RFA, 2022].  This experiment significantly improves the robustness of our validation, and Fig. A further demonstrates our method’s effectiveness in complex FL scenarios. Our method outperforms or matches other baselines across all datasets (Breast, Diabetes, small-MNIST, small-CIFAR10) and conditions (No attack, MP noise, MP inverted gradient, DP label flipping, DP inverted loss), except for Inverted gradient attacks in small-MNIST, where Trimmed mean performs better (note that this is 1 out of 20 cases).
> Additionally, we tested the scalability of our framework with up to 200 clients. As shown in Fig. D, our method introduces 1 extra minute per round with 200 clients, compared to FedAvg. In contrast, the baseline Krum introduces over 15 minutes per round. This demonstrates the efficiency and scalability of our approach.
>
> _Generalization across different models._
>
> __Our method, while focused on neural networks (NNs) and traditional FL, theoretically applies to other ML models like decision trees, though it requires appropriate aggregation processes and counterfactual (CF) generation.__
> We focused our study on NNs and traditional FL, using differentiable models capable of producing a global model through parameter aggregation. However, extending our method to other ML models, such as decision trees, presents an interesting and challenging opportunity for future work due to their discrete model parameters (e.g., splitting rules and thresholds). Our proposed method is based not on the similarity between client model parameters, which can be problematic with such heterogeneous architectures, but on the client model’s performance and their respective decision-making process. Therefore, in theory, our defense method (FBSs) of evaluating client behaviors still holds with these ML models but requires a suitable aggregation process and CF generation. Various aggregation methods have been proposed in the literature for such models, which can be adapted for this purpose [Wang et al., Decision Tree-Based Federated Learning, 2024]. Additionally, generating CFs from interpretable models, such as decision trees, is straightforward due to their inherently explainable nature.
>
> _Interpretability and Usability. “The paper, while clear in its technical explanations, may still be challenging for practitioners who are not experts”._
>
> To broaden the audience and facilitate the comprehension of our method, __we introduce the pseudocode of our algorithm (Algorithm 1) for creating the behavioral planes on the server and implementing our robust aggregation method (FBSs).__
>
> _Visualizations and Trajectories. Question: Can you provide more detailed examples or visualizations of client trajectories in the Error Behavioural Plane and Counterfactual Behavioural Plane (CBP)? Suggestion: Include more visual examples and explanations of client trajectories_
>
> In addition to the trajectories presented in Fig. 3 of the main paper, __we provided additional trajectories in the Appendix B (Fig. 8)__. Specifically, Fig. 8 illustrates client trajectories within the FBPs for different scenarios, including an Inverted Loss attack on the Synthetic dataset and a Data Flip attack on the Diabetes dataset, highlighting the distinct behavioral patterns of clients and attackers in these settings. These trajectories also demonstrate the possibility of identifying clusters of clients with similar data distributions, particularly on the CBP.
>
> _How does the method handle IID and non-IID data?_
> __Our method is robust in both IID and challenging non-IID scenarios, demonstrating higher accuracy in IID settings across most conditions.__
> Please note that in the main paper, we conducted all experiments in non-IID scenarios (Section 4.1), which are the most realistic yet challenging for robust aggregation methods, as even honest clients behave differently. Additionally, we compared the performance of our defense method under No-attack, Crafted-noise, Inverted-gradient, Label-flipping, and Inverted-loss attacks in both IID and non-IID settings. The table below shows that higher accuracy is achieved in the IID setting under almost all conditions compared to the non-IID setting, highlighting the increased complexity of operating in non-IID environments (Appendix B).
>
> | Cond.   | No attack | MP Noise | MP Grad | DP Flip   | DP I.Loss | Mean     |
> |---------|-----------|----------|---------|-----------|-----------|----------|
> | non-IID | 95.7±1.1  | 98.0±0.8 | 95.3±0.7 | 94.2±0.6  | 95.9±0.9  | 95.8±0.4 |
> | IID     | 98.2±0.3  | 98.4±0.4 | 98.2±0.2 | 96.4±0.9  | 93.7±1.0  | 97.0±0.4 |

---

> > ### Author Response · Authors · 2024-08-12
> >
> > Thank you for your valuable feedback. Please let us know if you have any further questions or if there are any points that need additional clarification.

---

### Author Rebuttal · Authors · 2024-08-06

# Answer to reviewers and ACs
We thank the reviewers for their insightful feedback. We are encouraged by their recognition of the novelty in our ideas and proposed methods for analyzing and visualizing client behaviors (eMJr, ULpn, aREF), and the significance of this work in the FL field (eMJr). We are pleased that our work was found to be clear, easy to follow (aREF) and grounded in a solid theoretical foundation (eMJr, ULpn). We appreciate ULpn’s acknowledgment of our method’s explanatory power, enhancing FL system interpretability. Reviewers’ feedback has certainly improved the quality of our manuscript and we hope we successfully addressed your concerns in this rebuttal and in our updated submission. We reply to shared questions here and address specific questions under each reviewer’s feedback.
# Summary of Changes
In response to the reviewers’ comments, we worked on improving the clarity and comprehensiveness of our paper with new sections and explanatory content. By incorporating a new dataset, a new SOTA baseline, and additional experiments, we have increased the robustness of our method’s validation. These improvements address the specific concerns and strengthen our work’s overall contribution. However, the core contributions and evaluations of our work remain unchanged. Our changes are summarized as follows:
- __Comprehensive Theoretical and Experimental Analysis of Computational Costs__ (Appendix A.6)
- __Expanded Experimental Validation:__ Validated on a more complex dataset, CIFAR10, in addition to the four previously used datasets, to further assess our method’s performance (Fig. A and Table in Appendix B)
- __Additional Baseline Comparison:__ Introduction of a recent robust aggregation baseline [K. Pillutla et al., RFA, 2022] in all experiments (Fig. A, new Table 5, 6, 7)
- __Attack Intensity Analysis:__ Verified robustness against varying attack intensities (Fig. E)
- __Behavioral Score Analysis:__ Evaluated the importance of information provided by our behavioral planes by recording the automatically extracted client behavioral scores from our defense method (Fig. B)
# Common Answers
_@Rev-eMJr,ULpn,aREF – Lack of computational cost analysis_

__We perform a comprehensive analysis of the computational cost of our methods. In the worst-case scenario, introducing counterfactuals (CFs) adds 7.6% more model parameters and 5.1% more GFLOPs for inference compared to the predictor alone. Overall our method adds 1 minute per round to FedAvg, while Krum adds over 15 minutes per round.__
The computational cost in FL frameworks includes local computation, communication overhead, and server-side computation

- __Local Computation.__ Our methods integrate a CF generator with the original predictor to explain decision-making processes and provide insights into client data distribution. For small NNs, local computation is minimal compared to communication latency and synchronization, allowing CF generation for predictor input without affecting training efficiency. As the predictor size increases, as shown with MNIST and CIFAR10, we can efficiently generate CFs for intermediate layers using a relatively small number of neurons. As shown in Table A, without losing performance, the generator requires only 2.7% of the predictor’s GFLOPs (ResNet-18) for inference on a 28x28 RGB image. Training time is proportional, as our generator trains end-to-end with the predictor.
- __Communication Overhead.__ The CF generator is transmitted with the predictor to the server for evaluation and aggregation. The increase in the number of transmitted parameters is marginal compared to the predictor, consisting of only 1.8% of the predictor’s parameters (Table A). This results in an additional 0.92 megabytes (MB) compared to the 49.68 MB required for the predictor.
- __Server-side Computation.__ This involves evaluating client models on a small validation set (e.g., 250 samples as shown in Fig. 9) and calculating pairwise distances between client CFs. Model evaluations, which required a single pass for each client model, are negligible compared to the computational load of calculating pairwise distances. We use the sliced Wasserstein distance with a complexity of $O(m \log m)$, where $m = n_{samples} \times 2=500$ [L148]. This operation is repeated for each unique pair of clients, leading to a total complexity of $O(n^2⋅m \log m)$, where $n$ is the number of clients. Compared to Krum’s complexity of $O(n^2⋅d)$, where $d$ is the number of model parameters, our method is more efficient for NNs with more than 4480 parameters, which is typical in practical applications
- __Overall Computational Cost.__ We compared the computational cost of our defense (FBSs) with Krum and FedAvg across different network dimensions and client numbers (Fig. D). For increasing client numbers, our method uses a CF generator with 7.6% of the predictor’s parameters (worst-case scenario) and 250 validation samples, as used in the paper. Our method scales better than Krum under both conditions, adding only 1 extra minute per round with 200 clients compared to FedAvg, while Krum adds over 15 minutes per round. This demonstrates its efficiency in large-scale scenarios

_@Rev-ULpn,aREF – Why does the CF generator affect predictor performance?_

__The CF generator affects the performance of the predictor because it is jointly optimized with the predictor and this is needed to generate CFs during training.__ Therefore training the CF generator alongside the predictor influences the training process of the predictor. The small change in performance, shown in Table 1, can be attributed to the additional loss providing extra information to navigate the optimization space more efficiently, similar to regularization terms. On the contrary, using a post-hoc CF generator (which does not affect performances) would necessitate training a separate generator for each client at the end of every epoch, significantly increasing the computational cost

---

### Decision · Program_Chairs · 2024-09-25

**Decision:**

Accept (poster)

**Comment:**

The paper introduces a method to track FL clients' dynamic behavior.  There is good agreement among the reviewers on the major strengths of the paper and its novelty.